# Local Bow Shock Environment during Magnetosheath Jet Formation: Results from a Hybrid-Vlasov Simulation

**Jonas Suni**[1], **Minna Palmroth**[1,2], **Lucile Turc**[1], **Markus Battarbee**[1], **Giulia Cozzani**[1], **Maxime Dubart**[1], **Urs Ganse**[1], **Harriet George**[3], **Evgeny Gordeev**[1], **Konstantinos Papadakis**[1], **Yann Pfau-Kempf**[1], **Vertti Tarvus**[1], **Fasil Tesema**[1], **and Hongyang Zhou**[1]

[1]Department of Physics, University of Helsinki, Helsinki, Finland
[2]Space and Earth Observation Centre, Finnish Meteorological Institute, Helsinki, Finland
[3]Laboratory for Atmospheric and Space Physics, University of Colorado Boulder, Colorado, USA

**Correspondence:** Jonas Suni (jonas.suni@helsinki.fi)

**Abstract.**

Magnetosheath jets are plasma structures that are characterised by enhanced dynamic pressure and/or plasma velocity. In this study, we investigate the formation of magnetosheath jets in four two-dimensional simulation runs of the global magnetospheric hybrid-Vlasov model Vlasiator. We focus on jets whose origins were not clearly determined in a previous study using the same simulations (Suni et al., 2021) to be associated with foreshock structures of enhanced dynamic pressure and magnetic field. We find that these jets can be divided into two categories based on their direction of propagation, either predominantly antisunward or predominantly toward the flanks of the magnetosphere. As antisunward-propagating jets can potentially impact the magnetopause and have effects on the magnetosphere, understanding which foreshock and bow shock phenomena are associated with them is important. The antisunward-propagating jets have properties indistinguishable from those of the jets found in the previous study. This indicates that the antisunward jets investigated in this paper belong to the same continuum as the jets previously found to be caused by foreshock structures, however, due to the criteria used in the previous study, they did not appear in this category before. These jets together make up 86% of all jets in this study. The flankward-propagating jets make up 14% of all jets and are different, showing no clear association with foreshock structures and exhibiting temperature anisotropy unlike the other jets. We suggest that they could consist of quasi-perpendicular magnetosheath plasma, indicating that these jets could be associated with local turning of the shock geometry from quasi-parallel to quasi-perpendicular. This turning could be due to bow shock reformation at the oblique shock caused by foreshock ULF wave activity.

## 1 Introduction

When the supermagnetosonic solar wind interacts with Earth's magnetic field, a bow shock forms ahead of the Earth's magnetic domain. The part of the shock where the interplanetary magnetic field (IMF) is roughly parallel to the shock normal direction is called the quasi-parallel bow shock. The part where the IMF is roughly perpendicular to the shock normal direction is known as the quasi-perpendicular bow shock. Earthward of the IMF field line tangential to the bow shock, solar wind particles can be reflected by the bow shock and travel back upstream along magnetic field lines and interact with the pristine solar wind, causing a foreshock to form (e.g. Eastwood et al., 2005b; Wilson, 2016). The part of the foreshock containing reflected electrons is called the electron foreshock, and its sunward edge is close to the tangential field line. The edge of the ion foreshock, which contains field-aligned electron and ion beams but exhibits no wave activity, is earthward of the electron foreshock edge. The interaction between the solar wind and the reflected ions generates Ultra Low Frequency (ULF) waves via the ion-ion beam right-hand instability (Gary, 1991). The waves are advected back toward the bow shock by the solar wind flow. The part of the foreshock containing these ULF waves along with suprathermal electrons and ions is known as the ULF foreshock (Eastwood et al., 2005b; Andrés et al., 2015), and because the wave generation requires

a finite time dictated by the instability growth rate (Blanco-Cano et al., 2009), the edge of the ULF foreshock (the ULF foreshock boundary) is earthward of the ion foreshock edge.

As the solar wind plasma traverses the bow shock, it is compressed, heated, and decelerated. The region of space where this shocked plasma flows around Earth's magnetosphere is known as Earth's magnetosheath (e.g. Lucek et al., 2005). The boundary between the magnetosheath and the magnetosphere is called the magnetopause. As with the bow shock, the magnetosheath can also be divided into two sub-regions. The part of the magnetosheath downstream of the quasi-parallel bow shock is known as the quasi-parallel magnetosheath. The ULF waves generated in the foreshock can be transmitted through the bow shock into the quasi-parallel magnetosheath, causing it to also be dynamic (e.g. Dimmock et al., 2014; Turc et al., 2023). The part of the magnetosheath downstream of the quasi-perpendicular shock is called the quasi-perpendicular magnetosheath.

The dynamic quasi-parallel magnetosheath exhibits many kinds of transient phenomena (Zhang et al., 2022). One such phenomenon is magnetosheath jets. They were first observed by spacecraft in 1996 (Němeček et al., 1998) and have since then been described in many observational studies (e.g. Savin et al., 2008; Amata et al., 2011; Dmitriev and Suvorova, 2015; Hietala and Plaschke, 2013; Plaschke et al., 2017, 2020; Gunell et al., 2014; Gutynska et al., 2015; Plaschke and Hietala, 2018; Wang et al., 2018; Goncharov et al., 2020; Raptis et al., 2022b) as well as in simulations (e.g. Karimabadi et al., 2014; Hao et al., 2016; Omidi et al., 2016; Palmroth et al., 2018b; Omelchenko et al., 2021). Magnetosheath jets are usually defined as structures or regions of enhanced dynamic pressure $P_{\mathrm{dyn}} = \rho_m v^2$, where $\rho_m$ is the mass density and $v$ the bulk speed of the plasma, in the magnetosheath, although specific definitions and terminology differ from study to study (see Plaschke et al., 2018). The "transient flux enhancements" studied by Němeček et al. (1998) were defined as enhancements of ion flux in the magnetosheath, while Hietala et al. (2012) defined "supermagnetosonic jets" as regions where the magnetosheath flow is supermagnetosonic. Karlsson et al. (2012, 2015) studied "plasmoids," regions of enhanced magnetosheath density, while Plaschke et al. (2013) defined "high speed jets" using the enhancement of $x$-directional dynamic pressure. In this study we employ the definition of Archer and Horbury (2013), whose "dynamic pressure enhancements" are defined as regions where the dynamic pressure in the magnetosheath is at least twice the time-average of the magnetosheath dynamic pressure. This definition was deemed to be most appropriate for capturing transient dynamic pressure enhancements in previous studies using the same simulation data as this study (Palmroth et al., 2021; Suni et al., 2021).

Magnetosheath jets occur mainly in the quasi-parallel magnetosheath (Plaschke et al., 2013; Archer and Horbury, 2013; Vuorinen et al., 2019), and they form particularly frequently and travel deeper into the magnetosheath when the angle between the IMF direction and the Sun-Earth line (the cone angle) is small, the solar wind Alfvén Mach number is high, and the solar wind density is low (LaMoury et al., 2021; Koller et al., 2023). The dynamic pressure enhancements of jets may be associated with either increased density, increased velocity, or both (e.g. Archer and Horbury, 2013). Jets are also usually associated with enhanced magnetic field strength (especially when the density is enhanced, see Plaschke et al., 2013; Archer and Horbury, 2013; Karlsson et al., 2015) and decreased plasma temperature (Archer et al., 2012; Dmitriev and Suvorova, 2012; Plaschke et al., 2013). Jets have typical spatial scales of 1 $R_{\mathrm{E}}$, but studies of jet morphology have found that the shapes and sizes of jets can vary quite significantly (see Plaschke et al., 2018). Jets whose propagation velocities are more aligned with the Sun-Earth line than the ambient magnetosheath flow velocity is, or form very close to the subsolar point and are advected by the magnetosheath flow, can reach the magnetopause. These jets have been found to be quite common, with magnetopause impacts being estimated to occur several times per hour (Plaschke et al., 2016). Jets impacting the magnetopause can have effects on the magnetosphere by e.g. launching magnetospheric ULF waves (Archer et al., 2013; Wang et al., 2022), causing reconnection at the magnetopause (Hietala et al., 2018), and enhancing particle precipitation into the ionosphere (Hietala et al., 2012).

Many different mechanisms for the formation of magnetosheath jets have been suggested. Hietala et al. (2009, 2012) proposed that ripples on the bow shock surface can allow solar wind plasma to traverse the shock with only minimal deceleration, while Archer et al. (2012) suggested that solar wind discontinuities passing through the bow shock could lead to dynamic pressure enhancement. According to Savin et al. (2012), hot flow anomalies (HFAs) at the shock could generate jets. Karlsson et al. (2015) proposed that foreshock short, large-amplitude magnetic structures (SLAMS) impacting the shock could travel through the shock and become jets. Raptis et al. (2022b), using MMS data, observed the formation of magnetosheath jets as a direct consequence of foreshock wave evolution and bow shock reformation by compressive structures.

Many kinds of compressive structures have been observed in the foreshock. Some of the strongest structures are associated with the ULF wave field, and they are traditionally separated into two main categories. Shocklets (Hoppe et al., 1981) resemble steepened ULF wave trains or small shocks, exhibit magnetic field enhancements $< 2$ times the IMF strength, and have scale sizes on the order of 1 $R_{\mathrm{E}}$. SLAMS (Schwartz and Burgess, 1991), on the other hand, appear to be isolated coherent structures exhibiting magnetic field enhancements between 3 and 5 times the IMF strength, and having smaller spatial extents than shocklets (Lucek et al., 2002, 2004, 2008). Schwartz and Burgess (1991) suggested that as SLAMS are advected by the solar wind flow toward the shock, their amplitudes grow until they resem-

ble magnetosheath plasma, at which point the SLAMS can merge with the bow shock in a process known as bow shock reformation.

Investigating the formation mechanism of a jet with spacecraft measurements is challenging, as observing jet formation requires very fortuitous conjunctions of multiple spacecraft (as in e.g. Raptis et al., 2022b). Numerical simulations do not have this limitation and are thus useful in investigating jet formation mechanisms. For instance, Omelchenko et al. (2021) used the 3D hybrid-PIC simulation HYPERS to formulate a theory that turbulent entanglement of solar wind and magnetospheric magnetic field lines can provide favourable conditions for incursion of fast solar wind plasma into the magnetosheath and the formation of magnetosheath jets.

In this study, we use the global hybrid-Vlasov simulation Vlasiator (Palmroth et al., 2018a) to investigate magnetosheath jets. Vlasiator has been found to accurately model foreshock processes (Palmroth et al., 2015; Turc et al., 2023), SLAMS and bow shock reformation (Johlander et al., 2022), and magnetosheath jets (Palmroth et al., 2021), showing agreement with spacecraft observations. In Suni et al. (2021), we investigated the formation of magnetosheath jets in four two-dimensional Vlasiator simulation runs. We found that under steady solar wind conditions and quasi-radial IMF, the formation of up to 75% of jets can be explained by foreshock structures of enhanced dynamic pressure and magnetic field impacting the bow shock. We called these structures "foreshock compressive structures" (FCS), defined as regions upstream of the bow shock that fulfill

$$P_{\mathrm{dyn}} \geq 1.2 P_{\mathrm{dyn,sw}} \tag{1}$$
$$|\boldsymbol{B}| \geq \eta |\boldsymbol{B}_{\mathrm{IMF}}|,$$

where $P_{\mathrm{dyn}}$ is the local dynamic pressure, $P_{\mathrm{dyn,sw}}$ is the solar wind dynamic pressure, $|\boldsymbol{B}|$ is the magnetic field magnitude, $|\boldsymbol{B}_{\mathrm{IMF}}|$ is the IMF magnitude, and $\eta$ is a threshold that can take values between 1.1 and 3.0. This encompasses, but is not limited to, the definition of shocklets and the lower bound of SLAMS, and makes no assumptions about any particular formation mechanism for the foreshock structures. In Suni et al. (2021), approximately 75% of the jets are associated with such structures for $\eta = 1.1$. We also found that the jets associated with FCSs (called FCS-jets) penetrate deeper into the magnetosheath than the other jets (called non-FCS-jets). The formation mechanisms of the remaining 25% of jets were left to a future study.

In this study, we investigate the 25% of jets not studied by Suni et al. (2021). Using statistical analysis, we compare them to the 75% of jets found to be connected to FCSs. We analyse the plasma and magnetic field properties at and around the formation time and location of the jets. As in Suni et al. (2021), we require that the jets form at the bow shock. We find that the 25% of jets propagate either predominantly antisunward or flankward. We separate the jets under study here into two classes based on propagation direction. Using case studies and statistical analysis, we compare the properties of these two classes to each other.

## 2 Model and methods

### 2.1 Vlasiator

Vlasiator (Palmroth et al., 2018a) is a global magnetospheric, high performance hybrid-Vlasov simulation. It models protons as velocity distribution functions and electrons as a massless charge-neutralising fluid. The proton distribution functions evolve in time according to the Vlasov equation, while the electromagnetic fields evolve according to Maxwell's equations. The plasma and fields are coupled through the generalised Ohm's law including the Hall current term. Vlasiator is intrinsically 6-dimensional (6D), with 3 position space dimensions $(x, y, z)$ and 3 velocity space dimensions $(v_x, v_y, v_z)$.

In this study we investigate four Vlasiator simulation runs (see simulation parameters in Table 1). These runs are the same ones studied by Palmroth et al. (2021) and Suni et al. (2021), and they neglect the position space $z$-dimension in order to limit the computational costs of the simulation. The four runs thus simulate the Geocentric Solar Ecliptic (GSE) $xy$-plane, with simulation domains large enough to capture the solar wind, foreshock, dayside magnetosheath and magnetosphere, and partially the nightside. The radius of the Earth is $R_{\mathrm{E}} = 6.371 \cdot 10^6$ m. In runs HM30 and LM30, the domain size is $\sim [-7.9, 47]$ $R_{\mathrm{E}} \approx [-220, 1311]$ $d_{\mathrm{i}}$ in $x$, $\sim [-31, 31]$ $R_{\mathrm{E}} \approx [-875, 875]$ $d_{\mathrm{i}}$ in $y$, and $\sim [-0.018, 0.018]$ $R_{\mathrm{E}} \approx [-0.5, 0.5]$ $d_{\mathrm{i}}$ in $z$, and the mesh size $(x, y, z)$ is $1530 \times 1750 \times 1$ cells. In runs HM05 and LM05, the domain size is $\sim [-7.9, 64]$ $R_{\mathrm{E}} \approx [-400, 3234]$ $d_{\mathrm{i}}$ in $x$, $\sim [-31, 31]$ $R_{\mathrm{E}} \approx [-1590, 1590]$ $d_{\mathrm{i}}$ in $y$, and $\sim [-0.018, 0.018]$ $R_{\mathrm{E}} \approx [-0.9, 0.9]$ $d_{\mathrm{i}}$ in $z$, and the mesh size is $2000 \times 1750 \times 1$ cells. The point dipole that generates the geomagnetic field is positioned at the origin and is implemented with the Earth's dipole moment, $8.0 \times 10^{22}$ Am$^2$. The dipole moment is aligned with the GSE $z$-axis, so the GSE coordinate system is equivalent to the GSM coordinate system in this case. The magnetopause standoff distance is around 8 $R_{\mathrm{E}}$ in all runs. This is slightly different from the $\sim 10$ $R_{\mathrm{E}}$ expected in reality, and it is likely due to the 2D geometry of the simulation runs, as discussed in Palmroth et al. (2018b). Figure 1a) shows the dynamic pressure in the entire simulation domain in run HM05 at an example time $t = 489.5$ s. The IMF is quasi-radial ($\leq 30°$ IMF cone angle) in all runs. The outer simulation boundaries are periodic in the out-of-plane ($\pm z$) directions, the $\pm y$ and $-x$ boundaries apply homogeneous Neumann conditions, and the $+x$ boundary is set according to the constant solar wind parameters. In all four runs the inner simulation boundary consists of a perfect conductor at a radius of 5 $R_{\mathrm{E}}$ from the

origin, which is at the center of the Earth. The high solar wind velocity was chosen to facilitate quick development of the bow shock and magnetosheath in the simulation. The Alfvén Mach number, which is the most important parameter for realistic evolution of the plasma environment near Earth's bow shock, is however within the normal range of observations at Earth in all the runs (Winterhalter and Kivelson, 1988; Ma et al., 2020).

## 2.2   Jet identification and tracking

In order to identify, separate and track magnetosheath jets over time, we use the methods developed in Palmroth et al. (2021) and Suni et al. (2021). We search for jets in a search box which is chosen to focus on the subsolar magnetosheath in each run and for a tracking duration, limited by the simulation duration of each run, starting at 290 s (see Table 1). The extents of the search box in run HM05 are marked with black dotted lines in Fig. 1a. We define jets according to the criterion presented in Archer and Horbury (2013) as regions consisting of cells in the magnetosheath where the instantaneous dynamic pressure is at least twice the 3-minute moving time average of the dynamic pressure, $P_{\mathrm{dyn}} \geq 2\langle P_{\mathrm{dyn}}\rangle_{3\mathrm{min}}$. Due to the limited simulation durations of the runs used in this study, we use a 3-minute time average instead of the original 20-minute time average used by Archer and Horbury (2013). The regions fulfilling this criterion at one time step in run HM05 are delineated with green contours in Fig. 1b. Jets that are identified at only one time step are discarded, as their propagation cannot be calculated from tracking the jet. While the method we use can identify jets anywhere in the magnetosheath, we additionally require that the jets we study form at the bow shock, as Palmroth et al. (2021) proposed that regions fulfilling the jet criteria that form deeper in the magnetosheath are merely momentary dynamic pressure fluctuations in a low ambient dynamic pressure environment rather than jets. We define the bow shock in two different ways adapted from Battarbee et al. (2020): The boundary where the temperature of the core ion population (as discussed in Wilson et al., 2014) is 3 times the solar wind temperature, $T_{\mathrm{core}} = 3T_{\mathrm{sw}}$ (plasma heating); and the boundary where the x-directional magnetosonic Mach number is 1, $M_{\mathrm{ms},x} = v_x/\sqrt{v_{\mathrm{s}}^2 + v_{\mathrm{A}}^2} = 1$ (red and yellow contours, respectively, in Fig. 1). A third definition, the boundary where the ion density is twice the solar wind density, $n = 2n_{\mathrm{sw}}$ (plasma compression, blue contour in Fig. 1), is also shown for comparison, but because this threshold is often fulfilled within foreshock structures that contribute to shock reformation (Schwartz and Burgess, 1991; Johlander et al., 2022), it frequently misidentifies foreshock structures as magnetosheath plasma, and thus it is not used in jet categorisation or analysis. The magnetosheath is defined using the temperature criterion as the region of the simulation where $T_{\mathrm{core}} \geq 3T_{\mathrm{sw}}$. A jet is considered to form at the bow shock if the simulation cells in position space comprising the jet are in contact with either the temperature boundary or the Mach number boundary. Because we require information about whether magnetosheath jets are connected to foreshock structures or not, instantaneous values of $T_{\mathrm{core}}$ and $M_{\mathrm{MS},x}$ are used instead of time-averaging to acquire smooth and stable boundaries as done in e.g. Ng et al. (2022). The x-directional Mach number is a suitable proxy for the bow shock location only near the nose of the bow shock, but as we search for jets in a subregion of the dayside magnetosheath (see Table 1), the behaviour of the x-directional Mach number at the flanks is not an issue. We discard jets that exist for only a single simulation output time step, as well as jets that are found through visual inspection to clearly be the same structure as another jet in the data set or which form at a location that is found not to actually be at the bow shock.

The jets that form at the bow shock are initially separated into two categories as in Suni et al. (2021): Those that form in contact with FCSs, called FCS-jets (marked with red dots in Fig. 1b); and those that do not, whose origins are unclear and which are called non-FCS-jets (marked with a black dot in Fig. 1b). In Suni et al. (2021) we defined FCS as structures upstream of the bow shock (with the boundary defined as $T_{\mathrm{core}} = 3T_{\mathrm{sw}}$) fulfilling the criteria of eq. 1. In order to capture even the weakest FCS, in this study we use a magnetic threshold of $\eta = 1.1$. The regions fulfilling the FCS criteria are delineated with grey contours in Fig. 1b).

## 3   Results

### 3.1   Jet Classification

In order to study the propagation of jets in the magnetosheath, we define for each jet

- a formation time $t_0$
- a formation site $(x_0, y_0)$,

where $t_0$ is the earliest simulation time step at which the jet is identified and, to emphasise the parts of the jet with higher dynamic pressure, $(x_0, y_0)$ is a weighted mean of the cells comprising the jet,

$$(\bar{x}, \bar{y})(t) = \left( \frac{\sum\limits_{k \in cells(t)} w_k x_k}{\sum\limits_{k \in cells(t)} w_k}, \frac{\sum\limits_{k \in cells(t)} w_k y_k}{\sum\limits_{k \in cells(t)} w_k} \right) \qquad (2)$$

$$w_k = \frac{P_{\mathrm{dyn},k}}{\langle P_{\mathrm{dyn},k}\rangle_{3\mathrm{min}}} - 2,$$

at $t_0$, $(x_0, y_0) = (\bar{x}, \bar{y})(t_0)$, where the weights $w_k$ are a measure of how much the dynamic pressure of each cell exceeds the criterion used to define the jets. To calculate the propagation velocity of the dynamic pressure enhancement associated with the jet, we define a formation of three virtual spacecraft (VSC) in an equilateral triangle centered on

**Table 1.** Parameters of the different simulation runs used in the study. From left to right, the columns give the run identifier, IMF vector in GSE, IMF strength, IMF cone angle, solar wind number density, solar wind velocity, solar wind Alfvén Mach number, solar wind ion inertial length, the box in which jets were searched for ($x_{\min}$, $x_{\max}$, $y_{\min}$, $y_{\max}$), and jet tracking duration. For all runs, the solar wind temperature is 0.5 MK, the position space resolution is 227 km, and the velocity space resolution is 30 $\mathrm{kms}^{-1}$.

| Run | $\boldsymbol{B}_{\mathrm{IMF}}$ [nT] | $\lvert\boldsymbol{B}_{\mathrm{IMF}}\rvert$ [nT] | Cone angle [°] | $n$ [cm$^{-3}$] | $v_x$ [kms$^{-1}$] | $M_{\mathrm{A}}$ | $d_{\mathrm{i,sw}}$ [km] | Search box [$R_{\mathrm{E}}$] | Tracking duration [s] |
|---|---|---|---|---|---|---|---|---|---|
| HM30 | ( −4.3, 2.5, 0 ) | 5 | 30 | 1 | −750 | 6.9 | 227.7 | ( 6, 18, −8, 6 ) | 129.5 |
| HM05 | ( −5.0, 0.4, 0 ) | 5 | 5 | 3.3 | −600 | 10 | 125.4 | ( 6, 18, −6, 6 ) | 299.5 |
| LM30 | ( −8.7, 5.0, 0 ) | 10 | 30 | 1 | −750 | 3.4 | 227.7 | ( 6, 18, −8, 6 ) | 379.5 |
| LM05 | ( −10.0, 0.9, 0 ) | 10 | 5 | 3.3 | −600 | 5 | 125.4 | ( 6, 18, −6, 6 ) | 149.5 |

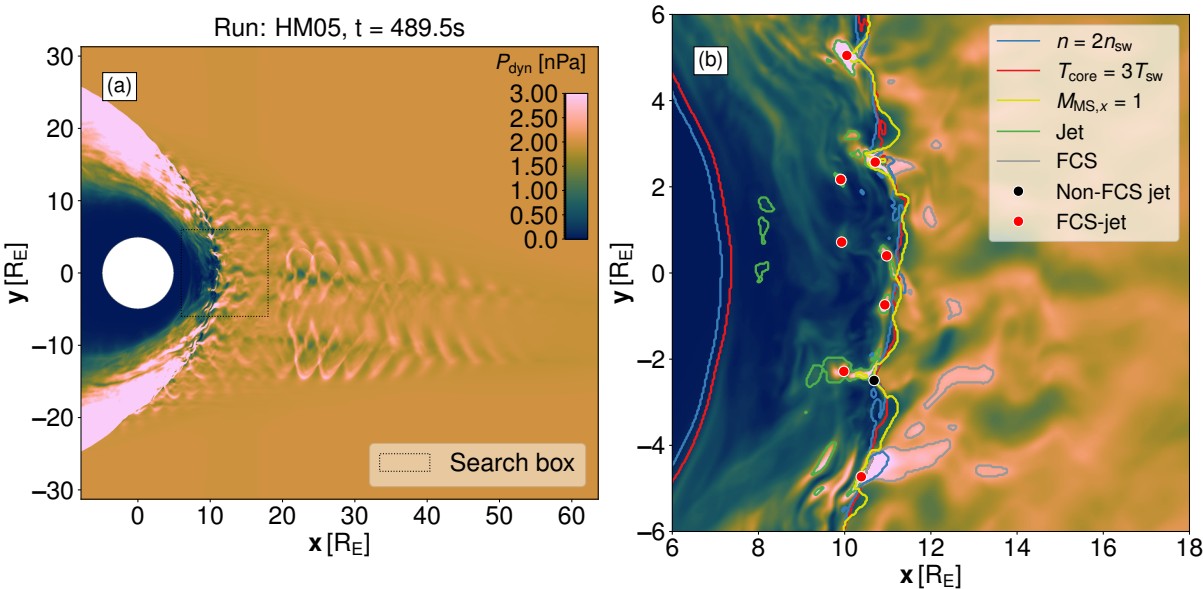

**Figure 1.** (a) Overview of dynamic pressure in the entire simulation box of run HM05 at an example time $t = 489.5$s. The dotted black box shows the extents of where we search for jets in this run. (b) Zoomed-in view of the search box at the same time in the same run. The plasma compression, plasma heating, and magnetosonic Mach number bow shock criteria are plotted as blue, red, and yellow contours, respectively. Jets and FCSs are delineated by green and grey contours, respectively. Non-FCS-jets and FCS-jets are marked with black and red dots, respectively.

$(x_0, y_0)$ and with an inter-spacecraft separation of $\sqrt{3}dx$, where $dx = 227$ km is the position space resolution (cell size) of the simulation runs (see Table 1 caption), which gives $\sqrt{3}dx \approx 393$ km $\approx 0.06$ $R_{\mathrm{E}}$ (see Fig 2a). Assuming that the propagating structure can be considered a plane wave, we apply multi-spacecraft timing analysis (Paschmann and Daly, 1998; Schwartz, 1998) to the time series of dynamic pressure measured from $t_0 - 10$ s to $t_0 + 10$ s at each of the three VSCs. The dynamic pressure time series of the reference spacecraft, for which the VSC at $(x_0, y_0 + dx)$ was chosen, is cross-correlated with the dynamic pressure time series of the other two VSCs, and the times of maximum cross-correlation are used to get time lags between the time series. Together with the VSC separations, this yields the propagation velocity $\boldsymbol{v_n}$ along the normal direction $\hat{\boldsymbol{n}}$ of the dynamic pressure enhancement corresponding to the jet. Be-

cause $v_n$ does not take into account the plasma bulk velocity perpendicular to the normal direction $\hat{\boldsymbol{n}}$, we estimate the total propagation velocity of the jet in the simulation frame as $\boldsymbol{v} = \boldsymbol{v_{\mathrm{bulk}}} + (v_n - \boldsymbol{v_{\mathrm{bulk}}} \cdot \hat{\boldsymbol{n}})\hat{\boldsymbol{n}}$ (similarly as in Archer et al., 2005), where $\boldsymbol{v_{\mathrm{bulk}}}$ is the mean bulk velocity measured by the reference spacecraft in the $[t_0 - 10$ s, $t_0 + 10$ s] interval. Subintervals where the VSC is considered to be in the foreshock (defined as $T_{\mathrm{core}} < 3T_{\mathrm{sw}}$), if any, are excluded.

For each jet, we also use the time-evolution of $(\bar{x}, \bar{y})$ to calculate an alternative propagation velocity $\boldsymbol{v_{\mathrm{tr}}}$, defined by the change of $(\bar{x}, \bar{y})$ from $t_0$ to $t_1$:

$$\boldsymbol{v_{\mathrm{tr}}} = \frac{(\bar{x}, \bar{y})(t_1) - (\bar{x}, \bar{y})(t_0)}{t_1 - t_0}, \tag{3}$$

where $t_1$ is taken 2 seconds (4 output time steps at 0.5 s time resolution) after jet formation. If the jet only exists for less

than 2 seconds, then $t_1$ is the last time when the jet is identified.

Because the calculation of $v$ is based on the observation of a temporal structure by a small number of virtual spacecraft while the calculation of $v_{tr}$ is based on the motion of a spatial structure consisting of cells in position space, it is expected that there will be some differences between the two estimates of the jet propagation velocity. For jets consisting of dynamic pressure enhancements that are large enough to be observed by all three VSCs and whose shapes do not change significantly during propagation, $v$ and $v_{tr}$ are expected to be quite similar. Because we do observe jets that are very small at the time of formation and then elongate in some direction, we expect there to be differences between $v$ and $v_{tr}$ in many cases.

Analysing the propagation velocities of non-FCS-jets, we find that they can be classified based on whether they propagate antisunward or toward the flanks. Thus, we classify the non-FCS-jets according to their directions of propagation in the simulation frame $v$: Jets whose propagation velocity vector is within $45°$ from the antisunward ($-x$) direction are classified as "antisunward jets", while the remaining jets are classified as "flankward jets". In cases when the maximum cross-correlation of the dynamic pressure time series between any VSC pair is less than $0.8$ (Eastwood et al., 2005a), we deem the timing analysis possibly unreliable and perform the classification based on $v_{tr}$ instead. This can occur when not all virtual spacecraft observe the jet-associated dynamic pressure enhancement clearly. After the non-FCS-jets are categorised in this way, the formation sites and times of the jets are visually inspected and jets that are not actually connected with the bow shock (as defined by the core heating or magnetosonic Mach number criteria) or which are clearly part of the same structure as any previously identified jet are discarded. Table 2 shows the number of jets of each category found in each of the four simulation runs, as well as the number of FCS-jets for comparison and the ratios of the numbers of antisunward, flankward, total non-FCS-jets, and FCS-jets to the total number of all jets. FCS-jets make up 71% of all jets. Because we discard jets that exist for only one time step in this study, and because the previous estimate of 75% found by Suni et al. (2021) was based on a figure, there is a slight difference between this study and Suni et al. (2021). Non-FCS-jets make up the remaining 29% of jets, and roughly half of the non-FCS-jets are antisunward and half flankward. The ratio of antisunward jets to flankward jets is $> 1$ for the $5°$ IMF cone angle runs, and $< 1$ for the $30°$ cone angle runs.

Figure 2b)-d) shows the propagation velocities $v$ and $v_{tr}$ for all flankward jets, antisunward jets, and FCS-jets respectively, as well as the medians of the propagation velocities and average bulk velocity at the reference VSC, with the median Alfvén and magnetosonic speeds at the reference VSC for comparison. $v$ obtained from timing analysis where the maximum cross-correlation between any VSC pair is below $0.8$ are not plotted or included in median calculations. The

$v_x$- and $v_y$-axes are both cropped to $[-1.3v_{sw}, 1.3v_{sw}]$ as this allows the majority of the data points to be shown without obscuring the medians and Alfvén and magnetosonic speeds. 6 flankward jet, 8 antisunward jet, and 58 FCS-jet $v$ data points fall outside the axes limits, while 0, 1, and 20 flankward, antisunward and FCS-jet $v_{tr}$ data point fall outside the axes. We can see that in the case of flankward jets, propagation is biased toward the dusk flank, as is the bulk flow, which suggests that most flankward jets form on the duskward side of the subsolar point. For antisunward jets and FCS-jets, on the other hand, the propagation velocities and bulk flow are distributed almost equally on the dawn and dusk sides, while the bulk flow is slightly biased toward dawn for FCS-jets. This is consistent with the deflection of the antisunward solar wind flow by the shock around the subsolar point. Most flankward jets propagate faster flankward than the bulk flow, while the propagation of antisunward jets appears to be quite closely aligned with the bulk flow. For all jet types, we can see some cases where $v_{tr}$ has no $x$-directional component. This is likely due to those particular jets being very short-lived and extending only one cell in the $x$-direction, in which case the weighted center coordinate is effectively quantised due to the finite simulation cell size. Indeed, 12% of flankward jets, 5% of antisunward jets, and 2% of FCS-jets have a lifetime maximum size of only one cell, but excluding these jets does not change the results of the analyses conducted in this study.

### 3.2 Case studies

Having classified the non-FCS-jets, we investigate possible differences in the plasma and magnetic field properties surrounding the formation of jets of different categories by selecting one typical flankward jet (marked with stars in Fig. 2b) and one typical antisunward jet (marked with stars in Fig. 2c) as examples for individual analysis. Figure 3 shows the properties surrounding the formation of the example flankward jet: Panel a) shows the dynamic pressure around the formation site (marked by the crosshairs) at $t_0$, with contours delineating regions where the different bow shock criteria and the jet and FCS criteria are fulfilled. The black dot marks the weighted center $(\bar{x}, \bar{y})$ of the flankward jet in question. The streamlines show the magnetic field. Panel b) shows the time series from $t_0 - 10$ s to $t_0 + 10$ s of dynamic pressure along a line segment centered on $(x_0, y_0)$ and extending 20 cells ($\sim 0.71\ R_E$) in the $-x$ and $+x$ directions. Also shown are contours marking the $x$-coordinates of the three bow shock criteria at $y_0$ as a function of time, and crosshairs marking $(x_0, t_0)$. Panel c) shows the results of the timing analysis for the flankward jet under consideration. Panels d)-h) show the time series from $t_0 - 10$ s to $t_0 + 10$ s at the formation site of density; velocity $x$-component, magnitude of the $yz$-components and the total magnitude; dynamic pressure; magnitude of the magnetic field $x$-component, $yz$-components and the total field; and temperature components perpendicular and parallel to the magnetic field. The deci-

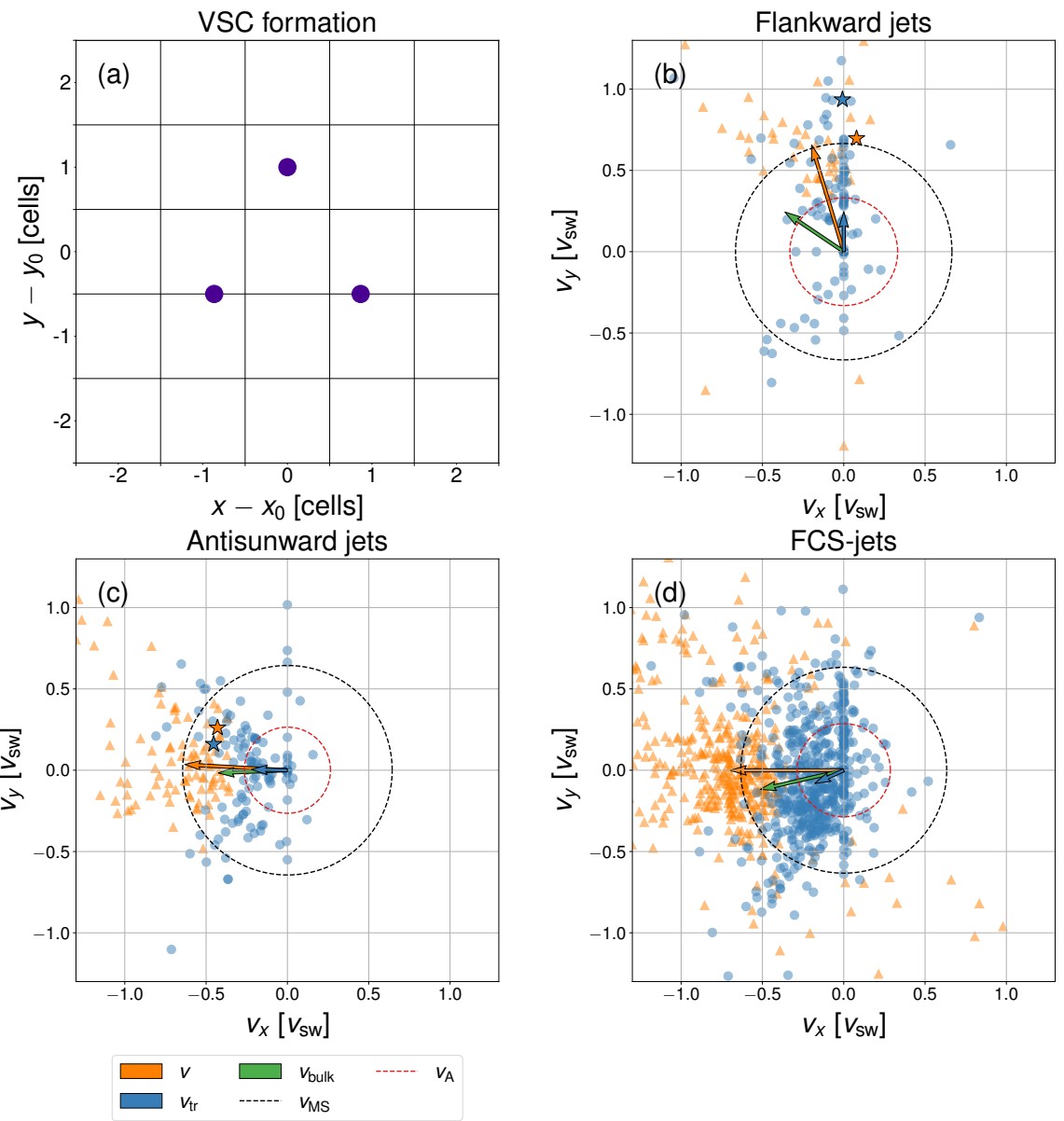

**Figure 2.** (a) Virtual spacecraft triangle formation, with respect to formation site $(x_0, y_0)$, used in the timing analysis. (b-d) Jet propagation velocities in the simulation frame calculated from timing analysis (orange triangles) and based on the tracking of the weighted centers of the jets (blue dots), as well as median bulk velocity (green arrow) and median Alfvén (red circle) and magnetosonic speeds (black circle) for (b) flankward jets, (c) antisunward jets, and (d) FCS-jets. Individual markers show the propagation velocity vectors of individual jets while the arrows show the medians of each velocity. The stars mark the propagation velocities of example jets used in the case studies. Jet propagation velocities from timing analysis for which the maximum cross-correlation between any VSC pair is less than $0.8$ are deemed unreliable and are not plotted or included in the calculations of the medians.

**Table 2.** Number of jets of each category and total number of non-FCS-jets found in each of the different simulation runs as well as in all runs combined. The number of FCS-jets found in each run and all runs combined are also given, as well as the proportions of the different jet categories and FCS-jets to the total number of jets.

| Run | Antisunward jets | Flankward jets | Total non-FCS-jets | FCS-jets | All jets |
|------|------|------|------|------|------|
| HM30 | 8 | 22 | 30 | 61 | 91 |
| HM05 | 31 | 15 | 46 | 145 | 191 |
| LM30 | 37 | 59 | 96 | 251 | 347 |
| LM05 | 44 | 13 | 57 | 105 | 162 |
| All | 120 | 109 | 229 | 562 | 791 |
| % of all jets | 15 | 14 | 29 | 71 | 100 |

sion to show $yz$-components instead of $y$ and $z$ separately was made to highlight the difference between the antisunward component and its orthogonal counterpart, as the magnetosheath flow and the effect of IMF clock angle on the magnetosheath are expected to be roughly rotationally symmetric. To improve clarity, we plot $|B_x|$ instead of $B_x$. as the sign of $B_x$ is mainly determined by the sign of the IMF $B_x$. The dashed lines mark $t_0$.

We can see from the simulation view (Fig. 3a) that the jet forms at the bow shock several $R_E$ duskward of the subsolar point, and that while waves are visible in the magnetic field on the upstream side, the dynamic pressure sunward of the jet is quite homogeneous, indicating the absence of compressional waves. This suggests that the formation site is close to the ULF foreshock edge. Immediately sunward of the formation site, the three bow shock criteria are not exactly colocated – the magnetosonic Mach boundary is sunward of the other two – i.e. the bow shock "non-local" (Battarbee et al., 2020). The cut-through time series shows that the formation of the jet is associated with lower dynamic pressure upstream of the formation site. Just before the formation time, the appearance of the bow shock non-locality can be seen. Fig. 3c) shows that the magnetosheath bulk velocity is sub-Alfvénic, while the jet propagation velocities are all super-Alfvénic in the simulation frame. The weighted center of the jet even propagates with supermagnetosonic speed. The time series show that the formation of the jet is associated with a large and steep increase in plasma density, as well as deflection of the plasma flow from $v_x$-dominated to $v_{yz}$-dominated. The formation is also preceded by a strengthening of $B_x$ and weakening of $B_{yz}$. Just before the formation time, there is a large increase in $T_\perp$ and a small decrease in $T_\parallel$, and as a consequence the temperature anisotropy $T_\perp/T_\parallel$ increases.

Figure 4 shows the surroundings of the forming example antisunward jet, presented in the same way as the example of the flankward jet. The simulation view in panel a) shows that this antisunward jet also forms several $R_E$ duskward of the subsolar point, but under very different conditions. As the simulation run in question has an IMF cone angle of $5°$, the formation site is downstream of the deep ULF foreshock. The magnetic field both upstream and downstream displays intense fluctuations, and compressional structures can be seen

on the upstream side. The bow shock immediately sunward of this jet is also non-local, but now the magnetosonic Mach number boundary is earthward of the other two bow shock boundaries. In the cut-through time series, we can see a foreshock dynamic pressure enhancement advecting toward and impacting the bow shock, which is followed by the formation of the jet at the impact location. A few seconds after the formation time, a bow shock reformation event (see Johlander et al., 2022) occurs, as shown by the blue contour extending further into the upstream in Fig. 4b). The timing analysis shows that the jet propagation velocities and ambient bulk velocity are all super-Alfvénic but submagnetosonic, and the jet propagates in the bulk flow direction. The time series show that the formation of the jet is associated with an increase in density and magnetic field strength, to which the $B_{yz}$ component contributes more than $B_x$. The formation is preceded by enhanced $v_x$ and a decrease in $v_{yz}$ as well as a decrease in both $T_\perp$ and $T_\parallel$ and approximately isotropic temperature.

### 3.3 Statistical analysis

The examples show that flankward and antisunward jets appear to differ in the properties of the plasma surrounding and comprising them. To investigate this further, we conduct a statistical study of all flankward, antisunward and FCS-jets. Figure 5 shows a superposed epoch analysis (SEA) of cut-through time series of plasma density (panels a, f, k), $v_x$ (panels b, g, l), dynamic pressure (panels c, h, m), magnetic field strength (panels d, i, n), and temperature (panels e, j, o), with $t_0$ of each jet serving as the epoch time and $x_0$ serving as the epoch $x$. Also shown are the average bow shock distances at $y_0$ as a function of time. We can see that on average, flankward jets are not associated with any foreshock structures convecting into the bow shock, at least not exactly sunward of the formation sites. Flankward jets also appear to be mainly density-driven, with little to no velocity enhancement in the magnetosheath at the formation site and time, but a noticeable enhancement of density. The plasma at the formation site and time is generally cooler than the ambient magnetosheath plasma but is surrounded by localised temperature enhancements. Flankward jets are associated with a sunward motion of the bow shock, with the motion becoming faster after the formation time. The jet formation is also followed

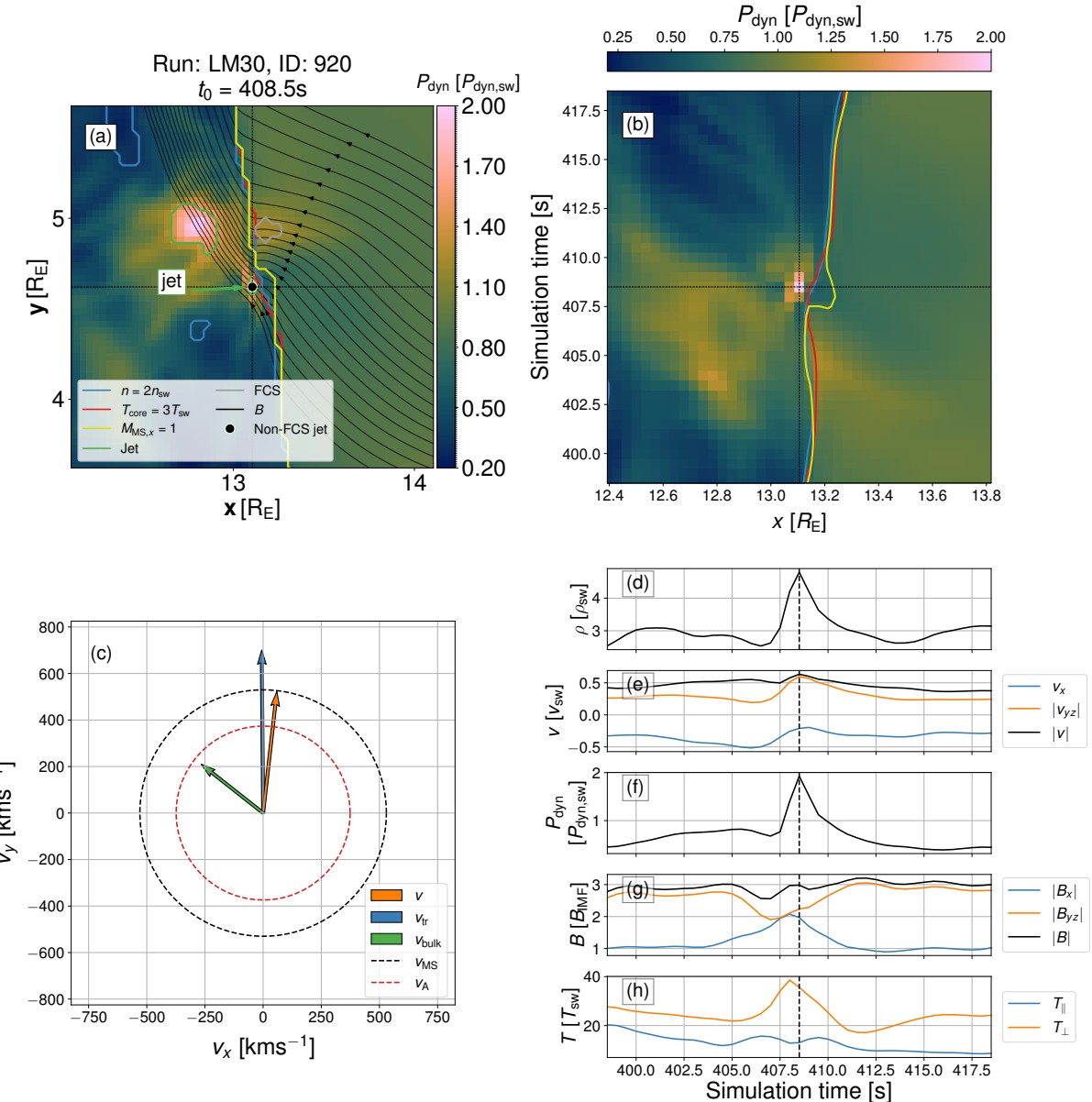

**Figure 3.** Properties of the near-bow shock environment around the formation time $t_0$ and place $(x_0, y_0)$ of an example flankward jet: (a) view of dynamic pressure with contours showing the fulfilling of the three bow shock criteria (plasma compression in blue plasma heating in red, and magnetosonic Mach number in yellow), the jet criteria (green), and FCS criteria (grey), with black dots indicating the weighted centers of tracked non-FCS-jets, and magnetic field lines shown as black streamlines; (b) cut-through time series around $t_0$ and $x_0$ at $y_0$ showing dynamic pressure and the three bow shock criteria as contours; (c) timing analysis from a triangle of VSCs centered on $(x_0, y_0)$, showing propagation velocities, bulk velocity, and Alfvén and magnetosonic speeds; (d-h) time series of plasma and magnetic field properties around $t_0$ at $(x_0, y_0)$.

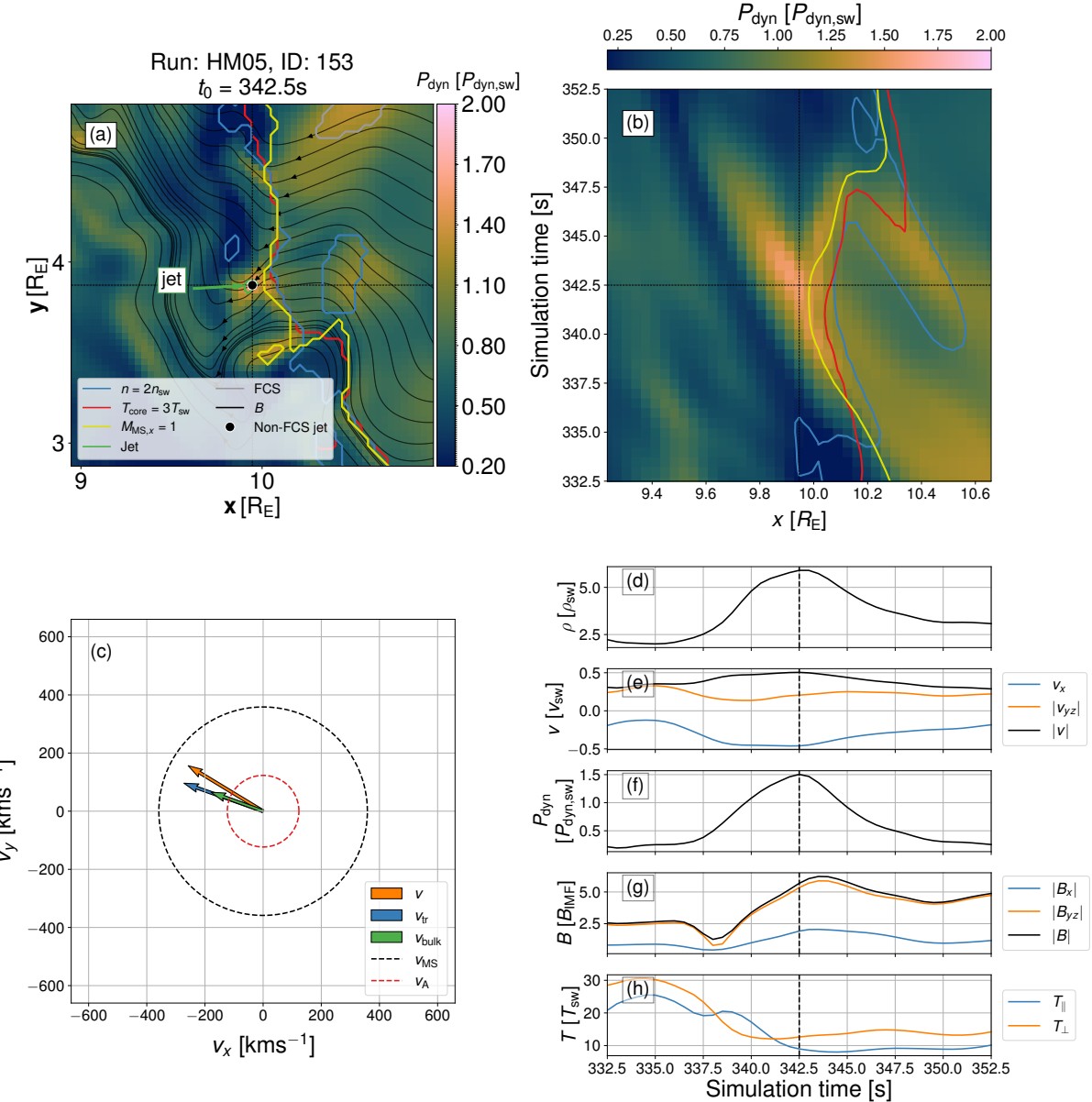

**Figure 4.** Same format as Figure 3 for an example antisunward jet.

by an enhancement of magnetic field strength in the magnetosheath. Antisunward and FCS-jets, on the other hand, appear to be very similar to each other and different from flankward jets. Both are associated with foreshock structures of enhanced density, dynamic pressure, and magnetic field advecting into the bow shock. The impact is concurrent with jet formation and the incursion of fast and cold solar wind plasma into the magnetosheath. For all jet types shown in Fig. 5, the bow shock location changes over time. The sunward motion occurring over the entire 30-second time window is most likely due to 2D simulation effects which cause the bow shock to move sunward in general, while the short time scale changes around the jet formation times could be local corrugation of the bow shock.

Figure 6 shows a SEA of the time series at the jet formation sites of plasma density (panels a, g, m), velocity components and magnitude (panels b, h, n), dynamic pressure (panels c, i, o), magnitude of magnetic field components and total field (panels d, j, p), parallel and perpendicular temperature (panels e, k, q), and temperature anisotropy (panels f, l, r) for flankward jets, antisunward jets, and FCS-jets. The formation time of each jet is chosen as the epoch time. Also shown are box-and-whisker plots showing the median, 25th percentile, 75th percentile, and the lowest and highest data points that are within 1.5 times the interquartile range (IQR) from the quartiles of the time series data used in the SEA sampled at epoch times $-6.5$ s, $0$ s, and $6.5$ s. We can see that flankward jets exhibit the greatest density enhancement on average, while FCS-jets exhibit the smallest. To some extent, this may be affected by the fact that most jets of all kinds are found in runs LM05 and LM30 (see Table 2), where the low solar wind Alfvén Mach number may lead to less compression of the plasma at the shock. In contrast, FCS-jets display the greatest enhancement in velocity, while flankward jets display the smallest. For flankward jets the velocity enhancement is in the $v_{yz}$-component, unlike for antisunward and FCS-jets where both $v_x$ and $v_{yz}$ are enhanced. The resulting enhancement in dynamic pressure is similar for all kinds of jets. The magnetic field enhancement is also similar across all categories, with the main difference being an enhancement in $B_{yz}$ after jet formation for flankward jets. The formation of all kinds of jets is preceded by and associated with a decrease in $T_\parallel$, but for flankward jets formation is associated with nearly constant $T_\perp$ that leads to enhanced temperature anisotropy, whereas for antisunward and FCS-jets $T_\perp$ also decreases around the formation time. The box-and-whisker plots show that there is considerable variation in the time series of jets belonging to each category, but the interquartile ranges are similar to the differences between the categories.

Finally, we investigate the formation sites of the different jet categories. Figure 7 shows the formation sites of flankward jets and antisunward jets, as well as FCS-jets for comparison, in the four simulation runs. Also depicted are the extent of the ion foreshock, defined as the presence of reflected ions (black dashed curve visible in panels b and d), the extent of the ULF foreshock, defined as enhancement of $B_z$, at $t = 400$ s in each run, similarly as in Turc et al. (2018) (pink and brown contours), and the extent of the jet search box (black dotted lines, see Table 1) in each run. The example jets studied in section 3.2 are marked with stars. We can see that antisunward and FCS-jets form everywhere at the bow shock in the search box in all simulation runs, but the majority of flankward jets form at the edge of the ULF foreshock on the dusk flank side, in agreement with the median bulk velocity in Fig. 2b, in the $30°$ IMF cone angle runs.

# 4 Discussion

In this study we have investigated the formation of jets that were not associated with foreshock compressive structures in Suni et al. (2021) by classifying them based on their direction of propagation. We have found that these non-FCS-jets can be separated into two categories based on their direction of propagation: Flankward and antisunward jets. We have conducted case studies by analysing two example jets in four different ways: 2D simulation views, cut-through time series, analysis of jet propagation, and virtual spacecraft time series at the formation site. We have performed a statistical analysis by conducting superposed epoch analyses of the cut-through time series and the virtual spacecraft time series, as well as compared the median propagation velocities and visualised where along the bow shock different jets form, for flankward jets and antisunward jets, as well as FCS-jets for comparison. We have found that antisunward jets have the same properties and origin as FCS-jets. Flankward jets, on the other hand, differ in many ways from the other jets.

As we have seen in Figure 5, antisunward jets and FCS-jets are both associated with foreshock structures of enhanced density, dynamic pressure and slightly enhanced magnetic field strength convecting into the bow shock, the impact coinciding with jet formation in the magnetosheath. The impact is also associated with the intrusion of fast and cold solar wind plasma into the magnetosheath. We find that the plasma in the jets is colder and faster than the surrounding plasma, which agrees with previous results from Vlasiator and Magnetospheric Multiscale (MMS) spacecraft observations (Palmroth et al., 2021) as well as data from the five Time History of Events and Macroscale Interactions during Substorms (THEMIS) spacecraft (Plaschke et al., 2013). From Figure 6 we have seen that the velocity enhancement at the jet formation site is mainly in $v_x$, the magnetic field enhancement is mainly in $B_x$, and the temperature decrease is both in the parallel and the perpendicular component, for both antisunward and FCS-jets. Figures 2 and 7 show that antisunward and FCS-jets form in the same regions of the magnetosheath and they both mainly propagate antisunward.

Our analysis of the antisunward jets thus indicates that antisunward jets were previously categorised in the non-FCS

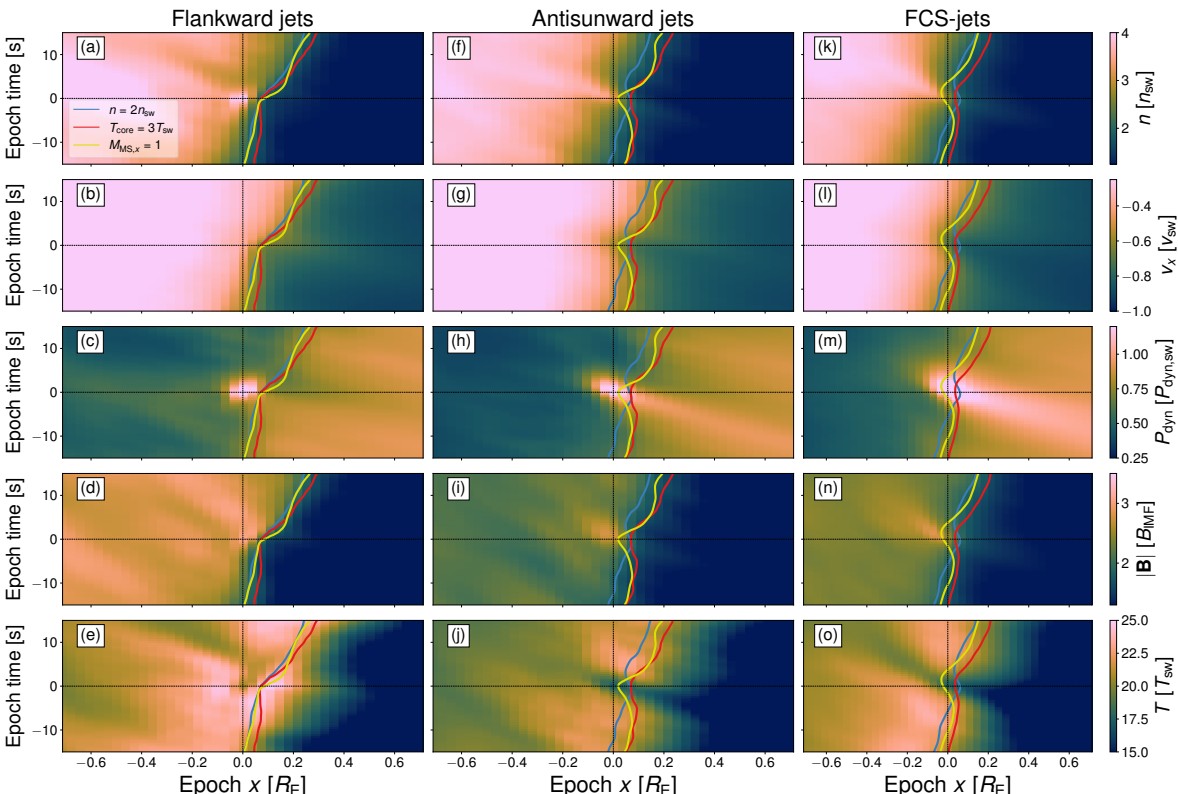

**Figure 5.** Superposed epoch analysis of cut-through time series of plasma density, $x$-directional velocity component, dynamic pressure, magnetic field strength, and temperature for flankward jets, antisunward jets, and FCS-jets. The formation time $t_0$ for each jet was chosen as the epoch time, while the formation site $x$-coordinate $x_0$ was chosen as the epoch location. Also shown are the average locations of the plasma density, plasma heating, and magnetosonic Mach number bow shock criteria as blue, red, and yellow contours, respectively.

group due to the selected thresholds for the parameter $\eta$ and/or the dynamic pressure threshold 1.2 in Eq. 1, and not because they are a fundamentally different phenomenon. Because antisunward and FCS-jets make up 683 of the total 790 jets used in this study (see Table 2), this means that 86% of jets that form at the bow shock under the steady solar wind conditions and quasi-radial IMF in our four simulation runs are associated with structures of enhanced dynamic pressure and magnetic field strength in the foreshock. As can be seen in Fig. 5, as the foreshock structures approach the bow shock, their density/dynamic pressure and magnetic field grow. This agrees with the formation mechanism of Raptis et al. (2022b), who used the MMS spacecraft constellation to observe foreshock waves becoming compressive when approaching the bow shock and subsequently causing bow shock reformation and magnetosheath jet formation as they make contact with the shock. While the average density and magnetic field in Fig. 5 do not quite reach magnetosheath values and thus systematic bow shock reformation is not visible, the steepening process of the structures/waves is similar. It should be noted, however, that the agreement between this study and Raptis et al. (2022b) does not exclude the possibility that the ripple formation mechanism proposed

by Hietala et al. (2009) or the magnetokinetic mechanism of Omelchenko et al. (2021) are also responsible for, or linked to, the formation of jets under similar or different solar wind conditions. However, we have not found these mechanisms at work in our simulations so far.

In contrast, the flankward jets that make up the remaining 14% of the jets investigated in this study have different properties. Their large temperature anisotropy (with $T_\perp$ being larger than $T_\parallel$, see Figure 6) and the deflection of the shocked solar wind plasma (enhancement of $v_{yz}$ rather than $v_x$) is reminiscent of the "quasi-perpendicular jets" described by Raptis et al. (2020). However, these quasi-perpendicular jets in Raptis et al. (2020) show modest enhancements or even decreases in the plasma density, while the flankward jets in this study are on average associated with higher density enhancements than antisunward and FCS-jets. The unexpectedly large overall density found in flankward jets could be due to the 2D nature of the simulation runs used in this study, as this prevents structures from dissipating in the out-of-plane direction (Pfau-Kempf et al., 2016; Suni et al., 2021), although this cannot account for the temporary enhancement in density. The observation that some jets are more density-driven than others is, however, consistent with spacecraft

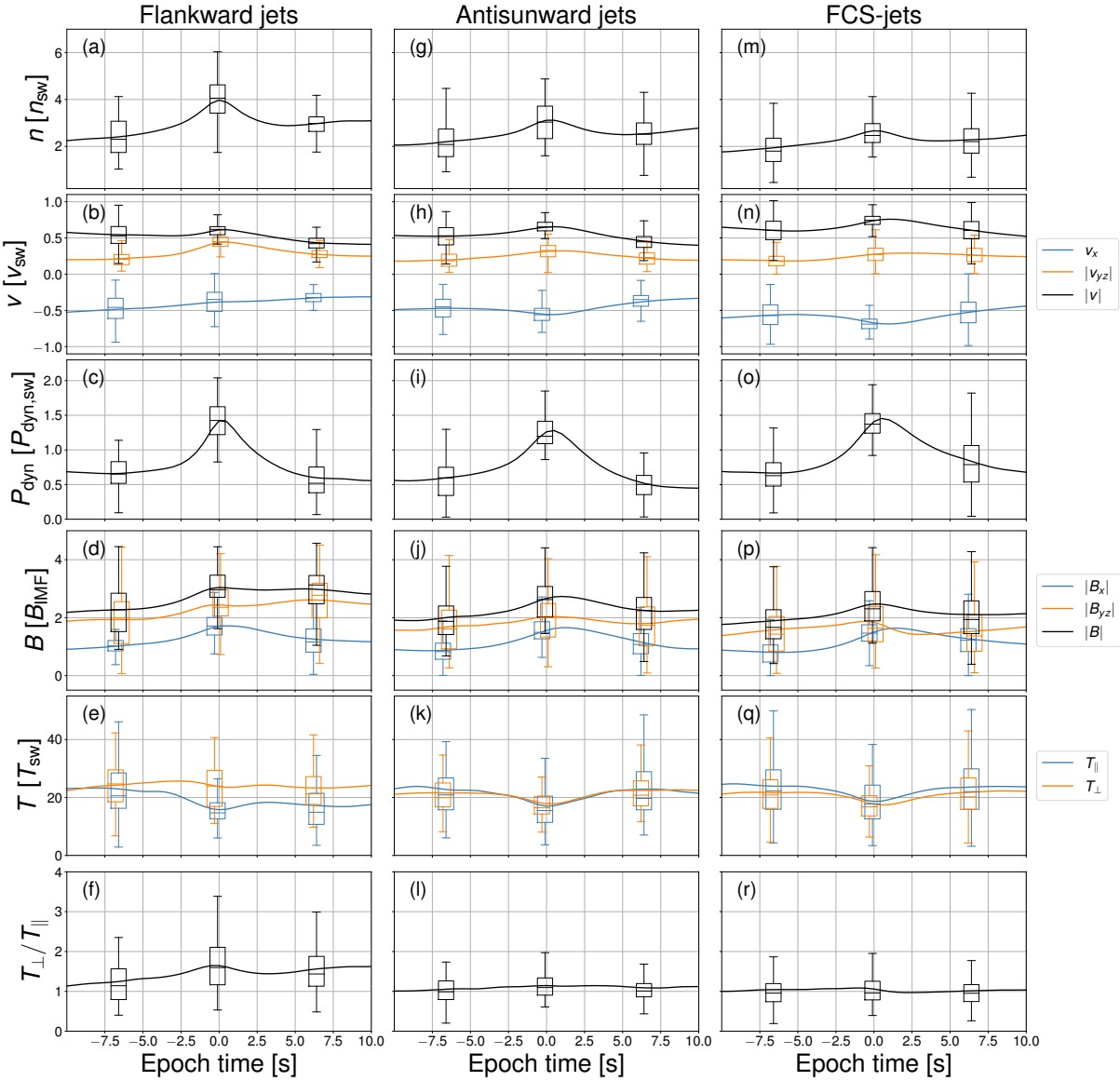

**Figure 6.** Superposed epoch analysis of single VSC time series, showing density, velocity components and magnitude, dynamic pressure, magnetic field components and magnitude, parallel and perpendicular temperature, and temperature anisotropy for flankward jets, antisunward jets, and FCS-jets. The formation time $t_0$ of each jet was chosen as the epoch time. Also shown are box-and-whisker plots at epoch times $-6.5$ s, $0$ s, and $6.5$ s. The boxes show the 25th percentile, the median and the 75th percentile, while the whiskers mark the lowest and highest data points that are within 1.5 times the interquartile range ($IQR = Q3 - Q1$) from the quartiles.

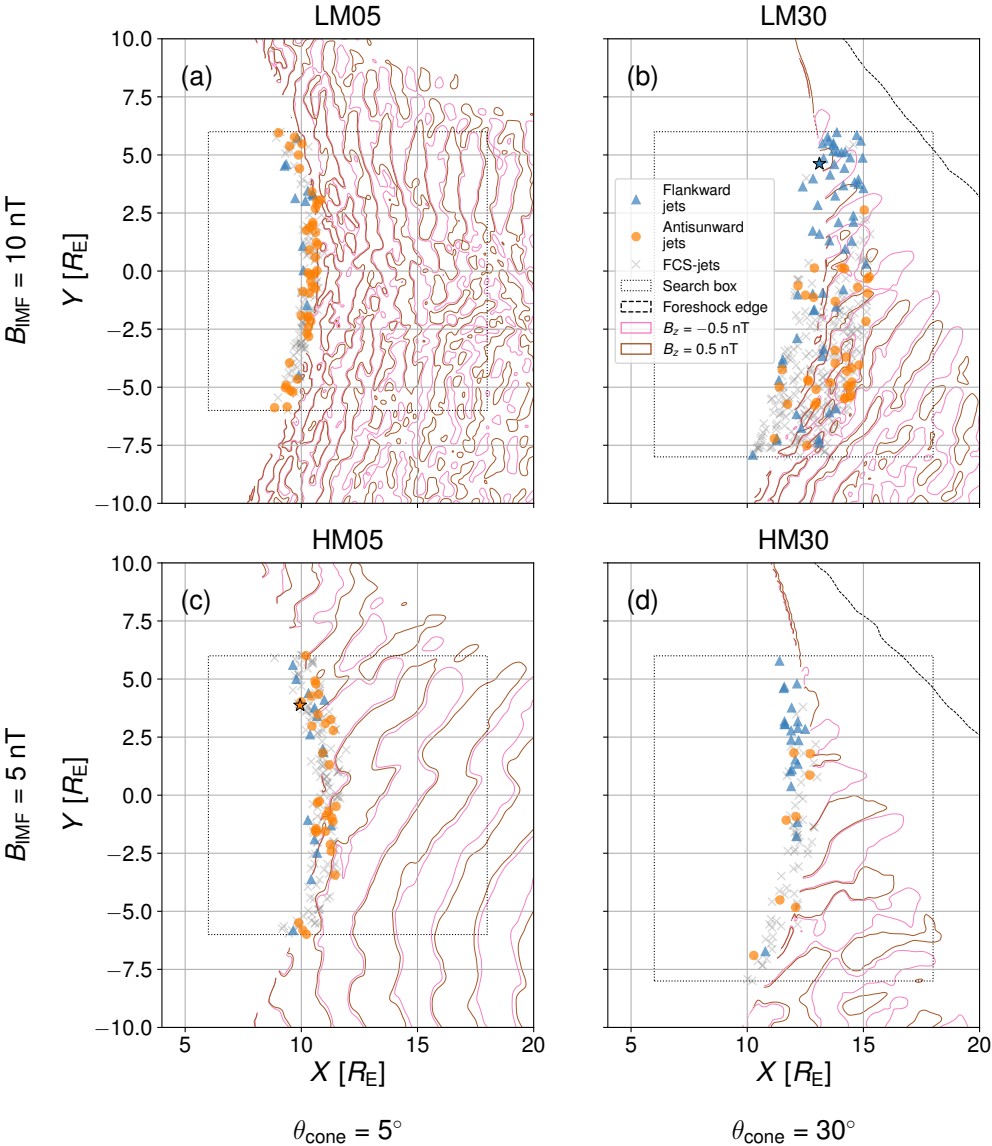

**Figure 7.** Formation sites $(x_0, y_0)$ of all non-FCS-jets separated by category, as well as FCS-jets for comparison, in the four runs: (a) LM05, (b) LM30, (c) HM05, and (d) HM30. Flankward and antisunward jets are shown as blue triangles and orange circles, respectively. The actual spatial extents of the jets are not shown. The jets used for the case studies are marked with stars. FCS-jets are shown as grey crosses. The dashed black curve shows the edge of the foreshock, as defined by the presence of reflected ions, and thus the extent of the ion foreshock at $t = 400$ s in each run. The pink and brown curves show, also at $t = 400$ s, the contours of $B_z = -0.5$nT and $B_z = 0.5$nT respectively, indicating the presence of ULF waves and the extent of the ULF foreshock. The horizontal dotted black lines show the extents of the jet search box in each run.

measurements (Archer and Horbury, 2013), and these could be related to the plasmoids of Karlsson et al. (2015). From Figure 6 we can also see that the temperature anisotropy at the formation site remains higher than 1 up to 10 seconds after jet formation, as does the enhancement of $B_{yz}$.

Our analysis of the flankward jets therefore suggests that the formation of flankward jets could be concurrent with a local change in bow shock geometry from quasi-parallel to quasi-perpendicular. Quasi-perpendicular shocks are associated with a sharper jump of plasma and magnetic field properties across the shock than quasi-parallel shocks, which could explain the enhancement of density inside the flankward jets. The enhancement of $v_{yz}$ and decrease in $v_x$ are also consistent with a change in the angle between the bow shock normal and the direction of the incoming solar wind. The local turning of a shock from quasi-parallel to quasi-perpendicular due to growing out-of-plane magnetic perturbations is a known phenomenon (e.g. Baumjohann and Treumann, 1996), but this turning has also been observed in association with bow shock reformation (Gingell et al., 2017; Liu et al., 2021). Liu et al. (2021) find that this turning can occur due to reformation at the oblique bow shock, which could explain why flankward jets form mainly at the edge of the ULF foreshock upstream of the oblique bow shock. At the oblique shock, the ULF waves are able to modulate the upstream conditions but do not generate compressional structures that would lead to the formation of antisunward or FCS-jets instead.

Considering the fact that the velocity enhancements of flankward jets are in the $y$-direction and that they are associated with weakening $v_x$ (see Fig. 6), it should be noted that these jets would not necessarily be captured if we had used a different jet criterion. For instance, the Plaschke et al. (2013) and Koller et al. (2022) criteria consider only $x$-directional dynamic pressure, which may not be enhanced in flankward jets. Furthermore, Raptis et al. (2022a) showed that the velocity distributions inside of magnetosheath jets can consist of multiple particle populations – a faster jet population and a slower background population, in which case using criteria that rely on plasma moments and derived properties, such as dynamic pressure, may not correctly identify all jets. The implications of this in Vlasiator will be investigated in a future study.

## 5 Conclusions

In this study we have investigated the origins of magnetosheath jets in four two-dimensional simulation runs of the global magnetospheric hybrid-Vlasov model Vlasiator. We have focused on jets that Suni et al. (2021) found not to be associated with foreshock compressive structures (FCS). We have found that these jets can be separated based on propagation direction, either antisunward or flankward. We have shown that the FCS-associated-jets previously investigated

in Suni et al. (2021) and the antisunward jets investigated in this study are fundamentally the same, and thus 86% of all the jets forming at the bow shock in four Vlasiator simulation runs with steady solar wind conditions and quasi-radial IMF form due to foreshock structures of enhanced dynamic pressure and magnetic field strength impacting the bow shock. Thus the reason why the antisunward jets were previously categorised in the non-FCS group is because the defining criteria of FCS-jets were too restrictive. The formation of these jets in the simulations is consistent with the formation mechanism observed with spacecraft in Raptis et al. (2022b).

We show that the remaining 14% of jets (the flankward jets) exhibit different properties from the 86% of jets that are associated with foreshock compressive structures. The flankward jets are not associated with foreshock structures and form mainly downstream of the ULF foreshock boundary. Instead, they display some features of quasi-perpendicular magnetosheath plasma, such as high temperature anisotropy and enhanced magnetic field and velocity in the direction transverse to the shock front. The flankward jets also have enhanced density, and in this respect they differ from spacecraft observations of jets in the quasi-perpendicular magnetosheath. These properties indicate that they might form behind a part of the bow shock that locally changes from quasi-parallel to quasi-perpendicular. If this is true, multi-spacecraft measurements should be searched for simultaneous observations of magnetosheath jets and signatures of the bow shock turning from quasi-parallel to quasi-perpendicular, such as disappearance of reflected ions just upstream of the shock. While jets that propagate flankward are not expected to have direct effects on the magnetosphere by impacting the magnetopause, these results advance our understanding of the effects that bow shock reformation can have in the magnetosheath.

To our knowledge, there are no observational studies characterising the propagation velocity of magnetosheath jets, to which we could directly compare our results. This is likely due to the challenges in obtaining the jet propagation velocity, which requires multi-point measurements. Future studies could for example revisit observations of magnetosheath jets from the four-spacecraft MMS mission to test whether observed jets can be categorised based on their propagation direction. Based on this study it would be also be important to search spacecraft observations for evidence of the local turning of the bow shock from quasi-parallel to quasi-perpendicular.

*Code and data availability.* Vlasiator is distributed under the GPL-2 open-source license. Vlasiator uses a data structure developed in-house. The Analysator software (Battarbee et al., 2021) was used to produce the presented figures. The runs described here can be either run with the above-mentioned code using the boundary conditions reported in this paper, or the data sets can be downloaded from the

University of Helsinki servers where they are stored (Pfau-Kempf et al., 2021b).

*Author contributions.* Conceptualisation: JS, MP. Data curation: JS. Formal analysis: JS. Funding acquisition: JS, MP, LT, YP. Investigation: JS, MP, LT, MB. Methodology: JS, MP, LT, MB. Project administration: MP. Resources: MP, YP, UG, MB. Software: MB, LT, YP, UG. Supervision: MP. Validation: JS, LT, MB. Visualisation: JS. Writing – original draft preparation: JS. Writing – review & editing: JS, LT, MB, MP, YP, HG, MD, FT, HZ, VT, GC, KP, EG.

*Competing interests.* At least one of the (co-)authors is a member of the editorial board of Annales Geophysicae. The peer-review process was guided by an independent editor, and the authors have also no other competing interests to declare.

*Acknowledgements.* The work of JS was made possible by a University of Helsinki funded doctoral researcher position in the Doctoral Programme in Particle Physics and Universe Sciences. We acknowledge the European Research Council for Starting grant 200141-QuESpace, with which Vlasiator (Pfau-Kempf et al., 2021a) was developed, and Consolidator grant 682068-PRESTISSIMO, awarded to further develop Vlasiator and use it for scientific investigations. We gratefully acknowledge the Academy of Finland grant nos. 336805, 328893, 322544, and 339756, and the Horizon 2020 FRoST grant no. 704681. The CSC – IT Center for Science in Finland and the PRACE Tier-0 supercomputer infrastructure in HLRS Stuttgart (grant nos. PRACE-2012061111 and PRACE-2014112573) are acknowledged as they made these results possible. The authors also wish to thank the Finnish Computing Competence Infrastructure (FCCI) for supporting this project with computational and data storage resources. LT acknowledges support from the Academy of Finland (grant number 322544) and from the University of Helsinki (three-year research project RESSAC 2020-2022). VT acknowledges support from the Academy of Finland (grant number 353197).

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
