# Peer review of "Local Bow Shock Environment during Magnetosheath Jet Formation: Results from a Hybrid-Vlasov Simulation"

_Annales Geophysicae, 2023_

## Referee Comment (RC1)

**The Local Bow Shock Environment during Magnetosheath Jet Formation: Results from a Hybrid-Vlasov Simulation**

**High speed jets in quasi-parallel shocks,** *submitted to Annales Geophysicae,*

The manuscript analyzes plasma structures identified by the authors in their previous simulations.

Major concerns:

1. In my opinion, the main question is: "What physical processes lead to the formation of jets in the simulations discussed?". The answer to this question remains unclear. In lines 299-301, the authors write: "The intrusion of solar wind plasma into the magnetosheath resembles what is expected by the SLAMS theory of jet formation by Karlsson et al. (2015), but here this applies to structures of dynamic pressure and magnetic field enhancement compared to the magnetic field enhancement that defines SLAMS." Karlsson et al (2015) do not present a "theory". Instead, they simply cite some SLAMS related computational works and Jovian magnetosheath observations. When Karlsson et al (2015) cite the work by Karimabadi et al. (2014), they say "SLAMS have been proposed to penetrate the bowshock and convect into the terrestrial magnetosheath during certain circumstances". This argument, however, is not directly linked to the formation of jets (dynamic pressure enhancements), observed by Karimabadi et al. (2014) in their 2D simulations and claimed to be caused by the ripple formation mechanism without providing arguments to support that claim. In fact, a more recent 3D computational study by Omelchenko et al. (2021) established a connection (cited in this manuscript) between the magnetosheath turbulence and formation of jets capable of penetrating the magnetosheath and impacting the magnetopause. Interestingly, their 3D jets look very similar in characteristics (density, magnetic field, speed, but not geometry) to the 2D jets demonstrated by Karimabadi et al (2014). Moreover, these jets do not seem to be related to SLAMS that normally originate in the foreshock, further away in the sunward direction. As far as the study by Karlsson et al (2015) is concerned, they simply hypothesize that "a small-amplitude foreshock SLAMS may encounter a corrugation in the bow shock (due to either a large-amplitude SLAMS or an HFA) and cross the bow shock". In this manuscript, the authors must support their conclusion that the nature of simulation jets "agrees with the formation mechanism proposed by Karlsson et al. (2015)" (line 308). For that, the authors must prove that these jets are in fact SLAMS, caused by the steepening of sunward propagating ULF waves – and not another type of plasma structures, e.g., ones that result from the turbulent action of magnetic field, as proposed by Omelchenko et al. (2015), whereby high-speed jets effectively represent "parcels" of upstream solar

wind plasma capable of deep penetration into the magnetosheath.  This extra analysis needed may be facilitated by comparing the characteristics of jets discussed in these simulations with those of SLAMS or high-speed jets (HSJs) observed experimentally. A mere classification of numerical "jets", carried out in this manuscript, is not sufficient for proving the ability of these simulations to reproduce data obtained in relevant satellite observations.

2. The dynamic pressure structures ("jets") shown in the Figures (e.g., Figs. 1,7) are "dot-like". It is unclear if the simulations resolve them at all.  How many computational cells do these "jets" spread over? The manuscript appears to claim that these structures are SLAMS. However, SLAMS are known to be characterized by large-amplitude magnetic perturbations (> 3 IMF strength) that initially propagate in the sunward direction in the foreshock. I do not see any proof of this mechanism of formation in the Figures presented. Do these simulations see SLAMS in the foreshock? In fact, these "dot-like" jets, if not properly resolved numerically, may be numerical artefacts, or LOCAL nonlinear effects associated with wave steepening/breaking (not to confuse with steepening of PROPAGATING wave fronts which may lead to formation of SLAMS).

3. The authors emphasize that the jets they study "are required to form at the bow shock" (line 124). For this purpose, they use three different ways to define the bow shock numerically (line 126). However, all these definitions use instantaneous simulation information. For instance, the authors state that they cannot use the density boundary because it fluctuates due to shock reformation (lines 133-134). In my opinion, using the other (temperature and Mach defined) boundaries as proxies for finding the shock location is not a safe physical approach for jet classification either. A more physically sound approach would be using time-averaged (slowly fluctuating) shock boundaries, e.g., similar to ones demonstrated by Ng et al. (JGR, Hybrid Simulations of the Cusp and Dayside Magnetosheath Dynamics Under Quasi-Radial Interplanetary Magnetic Fields, 2022). The authors should further discuss the uncertainty of their approach.

4. In the last paragraph of Section 2.1, the authors do not provide enough physical and numerical information about the setup of 2D Vlasiator simulations they analyze in this manuscript. At the very least, they should provide the (1) domain size in solar wind proton inertial lengths, (2) mesh resolution (numbers of cells in each dimension), and (3) simulation magnetopause to obstacle radius ratio (which characterizes the simulation dipole strength chosen).  For the reader to understand how realistic these simulations are, the authors should also discuss how they scale their ion inertial length scales to RE distances shown in the Figures.

5. Lines 160-165: It is not clear what the authors did here. VSCs are immobile points, aren't they? If so, their frame of reference coincides with the simulation frame of reference. Therefore, it is unclear what v_SC ("the propagation velocity in the spacecraft frame") represents in this numerical analysis. Does it make sense to measure it with respect to any point in the simulation (e.g., VSCs)?  Isn't it just v_n?  Please revise this paragraph for clarity.

6. Section 5 (Conclusions) needs to be revised in accordance with my previous remarks, especially those regarding "the formation mechanism proposed by Karlsson et al (2015)", who simply hypothesized about the origin of jets. There is no so evidence in this manuscript that the magnetosheath structures discussed are the SLAMS discussed by Karlsson et al (2015). The conclusion should also make clear which "jets" the authors refer to, given the nomenclature adopted in this manuscript (e.g., FCS-jets vs non-FCS jets). In regard to the simulation jets, the Conclusion mentions: "These properties indicate that they might form behind a part of the bow shock that locally and temporarily changes from quasi-parallel to quasi-perpendicular". This is not consistent with the SLAMS related explanation pointed out by Karlsson et al (2015). The statements in the Abstract and Conclusions sections must be clear and consistent.

Minor concerns:

1. The Abstract contains many insignificant details which obscure the main results of this work. The Abstract also contains information that must be supported by references, for instance: "A jet generation mechanism that has been widely discussed in observational and numerical studies is steepened Ultra Low Frequency (ULF) waves interacting with the bow shock. However, other formation mechanisms have also been proposed". For clarity, such statements should be avoided in the Abstract.
2. Line 44: replace "is host" by "is a host"
3. Line 82: replace "E.g." by "For instance,"
4. Line 87: replace "simulation runs" by "two-dimensional simulation runs" (it is important to emphasize 2D at the very beginning to avoid further confusion)
5. Last two sentences in Section 1. It is not clear what types of "jets" are being studied in this manuscript and what makes "non-FCS jets" different from "FCS jets". This must be discussed before the reader gets to see results from the statistical analysis. Also, this manuscript lacks comparisons with observations. That must be mentioned/explained.
6. Line 108: mention that you are using GSM axes
7. Line 129: provide a definition of "magnetosonic Mach number" to avoid confusion. Are you using the magnetosonic speed, $v_{ms} = \sqrt{v_a^2 + v_s^2}$ so that $M_{ms} = v_{sw}/v_{ms}$?
8. Line 131: rephrase "the position space simulation cells" (what does it mean?)

---

## Author Response (AR1)

**Author response**

The authors wish to thank the referees who reviewed the original manuscript. Their proposed changes and comments have significantly improved the quality and structure of the revised manuscript. In the revision, we have particularly focused on de-emphasising the jet formation mechanism hypothesised in Karlsson et al. [2015] and instead discussing jet formation in the context of Suni et al. [2021] and Raptis et al. [2022b]. We have also added more technical details about the simulations as well as referred to papers that validate Vlasiator with spacecraft data, expanded on the description of the timing analysis and how it compares to tracking spatial structures, and clarified why and how the different bow shock criteria are used.

This document consists of the author responses written to the referees before the revision of the manuscript, together with lists of changes made in the revised version. These lists are highlighted in **bold**.

**Author response to referee #1**

The authors wish to thank Anonymous referee #1 for their very insightful and thorough comments on the manuscript. We will take the comments into account when revising the manuscript. In this document we provide responses to each of the referee's comments (formatted as italics in indented paragraphs).
* * *
*In my opinion, the main question is: "What physical processes lead to the formation of jets in the simulations discussed?". The answer to this question remains unclear. In lines 299-301, the authors write: "The intrusion of solar wind plasma into the magnetosheath resembles what is expected by the SLAMS theory of jet formation by Karlsson et al. (2015), but here this applies to structures of dynamic pressure and magnetic field enhancement compared to the magnetic field enhancement that defines SLAMS." Karlsson et al (2015) do not present a "theory". Instead, they simply cite some SLAMS related computational works and Jovian magnetosheath observations. When Karlsson et al (2015) cite the work by Karimabadi et al. (2014), they say "SLAMS have been proposed to penetrate the bowshock and convect into the terrestrial magnetosheath during certain circumstances". This argument, however, is not directly linked to the formation of jets (dynamic pressure enhancements), observed by Karimabadi et al. (2014) in their 2D simulations and claimed to be caused by the ripple formation mechanism without providing arguments to support that claim. In fact, a more recent 3D computational study by Omelchenko et al. (2021) established a connection (cited in this manuscript) between the magnetosheath turbulence and formation of jets capable of penetrating the magnetosheath and impacting the magnetopause. Interestingly, their 3D jets look very similar in characteristics (density, magnetic field, speed, but not geometry) to the 2D jets demonstrated by Karimabadi et al (2014). Moreover, these jets do not seem to be related to SLAMS that normally originate in the foreshock, further away in the sunward direction. As far as the study by Karlsson et al (2015) is concerned, they simply hypothesize that "a small-amplitude foreshock SLAMS may encounter a corrugation in the bow shock (due to either a large-amplitude SLAMS or an HFA) and cross the bow shock". In this manuscript, the authors must support their conclusion that the nature of simulation jets "agrees with the formation mechanism proposed by Karlsson et al. (2015)" (line 308). For that, the authors must prove that these jets are in fact SLAMS, caused by the steepening of sunward propagating ULF waves – and not another type of plasma structures, e.g., ones that result from the turbulent action of magnetic field, as proposed by Omelchenko et al. (2015), whereby high-speed jets effectively represent "parcels" of upstream solar wind plasma capable of deep penetration into the magnetosheath. This extra analysis needed may be facilitated by comparing the characteristics of jets discussed in these simulations with those of SLAMS or high-speed jets (HSJs) observed experimentally. A mere classification of numerical "jets", carried out in this manuscript, is not sufficient for proving the ability of these simulations to reproduce data obtained in relevant satellite observations.*

It is true that Karlsson et al. [2015] simply make a hypothesis based on references to previous studies and observed similarities between paramagnetic plasmoids and SLAMS. Palmroth et al. [2018] and Raptis et al. [2020] found evidence in favour of the hypothesis in a simulation and in MMS spacecraft data, respectively. Based on this evidence, and the fact that Vlasiator simulations have been found to accurately model foreshock processes [Palmroth et al., 2015, Turc et al., 2023], SLAMS and bow shock reformation [Johlander et al., 2022], and magnetosheath jets [Palmroth et al., 2021] based on comparisons with spacecraft data, we further studied jets in Vlasiator simulations in Suni et al. [2021] and found that a majority of the jets were associated with foreshock structures of enhanced dynamic pressure and magnetic field. While the majority of the foreshock structures did not fulfill the criteria for SLAMS, some did fulfill the SLAMS criteria. The results of Suni et al. [2021] suggested that SLAMS and weaker foreshock magnetic field enhancements are part of a continuum of foreshock compressive structures of varying amplitudes, and thus we concluded that jets can form through a mechanism that is similar to the one hypothesised by Karlsson et al. [2015], even if the structures in question are not actually SLAMS. Raptis et al. [2022b] found direct spacecraft evidence for a connection between foreshock wave evolution, compressive foreshock structures, bow shock reformation, and the formation of magnetosheath jets. It should be stressed, however, that these results do not disprove either the bow shock ripple mechanism of Hietala et al. [2009] or the magnetokinetic mechanism of Omelchenko et al. [2021], they only show that the majority of jets in Vlasiator are caused by foreshock compressive structures interacting with the bow shock and magnetosheath.

The manuscript will be revised to focus the discussion about formation mechanisms of FCS-jets and antisunward jets on the results of Suni et al. [2021] and Raptis et al. [2022b]. In addition, we will refer to previous studies that have found agreement between Vlasiator and spacecraft observations.

- **We have revised the manuscript to focus on discussing the results of this study in the context of Suni et al. [2021] and Raptis et al. [2022b] instead of Karlsson et al. [2015]. We now also mention that we do not attempt to disprove the formation mechanisms of Hietala et al. [2009] or Omelchenko et al. [2021].**

- **We have added references to studies that found agreement between Vlasiator and spacecraft observations to the introduction.**
* * *
*The dynamic pressure structures ("jets") shown in the Figures (e.g., Figs. 1,7) are "dot- like". It is unclear if the simulations resolve them at all. How many computational cells do these "jets" spread over? The manuscript appears to claim that these structures are SLAMS. However, SLAMS are known to be characterized by large-amplitude magnetic perturbations (> 3 IMF strength) that initially propagate in the sunward direction in the foreshock. I do not see any proof of this mechanism of formation in the Figures presented. Do these simulations see SLAMS in the foreshock? In fact, these "dot-like" jets, if not properly resolved numerically, may be numerical artefacts, or LOCAL nonlinear effects associated with wave steepening/breaking (not to confuse with steepening of PROPAGATING wave fronts which may lead to formation of SLAMS).*

The black and red dots in Fig. 1 and the grey crosses, blue triangles and orange dots in Fig. 7 mark the weighted centers of jets as defined in Section 3.1, not their actual spatial extents. Approximately 12% of flankward jets, 5% of antisunward jets, and 2% of FCS-jets have a maximum size of only 1 cell (see Fig. 1 of this document). Excluding single-cell jets does not change the result of our analysis.

[Figure]

Figure 1: Histograms of maximum jet sizes in units of simulation cells divided by category. Jets larger than 20 cells exist as well, but the *x*-axis has been limited to the range [1,20] for better visibility of the distribution of small jets.

It was not our intent to imply that these structures, or any other structures in the magnetosheath, necessarily originate from SLAMS. The text as a whole will be revised to de-emphasise any perceived role of SLAMS in the formation of these simulated jets and to emphasise the role of foreshock compressive structures instead.

- **We have clarified that the markers in Figs. 1 and 7 mark the weighted centers of jets and not the actual spatial extents.**

- **We have confirmed that the results do not change if jets with a maximum size of 1 cell are excluded from the analysis, and mentioned this in the revised manuscript.**
* * *
*The authors emphasize that the jets they study "are required to form at the bow shock" (line 124). For this purpose, they use three different ways to define the bow shock numerically (line 126). However, all these definitions use instantaneous simulation information. For instance, the authors state that they cannot use the density boundary because it fluctuates due to shock reformation (lines 133-134). In my opinion, using the other (temperature and Mach defined) boundaries as proxies for finding the shock location is not a safe physical approach for jet classification either. A more physically sound approach would be using time-averaged (slowly fluctuating) shock boundaries, e.g., similar to ones demonstrated by Ng et al. (JGR, Hybrid Simulations of the Cusp and Dayside Magnetosheath Dynamics Under Quasi-Radial Interplanetary Magnetic Fields, 2022). The authors should further discuss the uncertainty of their approach.*

Both the method used in this study and the one in Ng et al. [2022] have merits and value, however, we want to study the transmission of transient structures through the bow shock and into the magnetosheath, so using instantaneous boundaries is necessary to correctly identify the contact between upstream and downstream structures. The reason that the density-based boundary is not used for this purpose is because the density of the upstream structures increases as they approach the bow shock, causing them to sometimes fulfill the density bow shock criterion and thus be counted as magnetosheath plasma according to the density criterion before they merge with the existing magnetosheath. This bow shock reformation has been studied in Vlasiator before [Johlander et al., 2022].

We will revise the manuscript to address this and compare and contrast our approach with that of Ng et al. [2022].

- **We have revised the Model and Methods section to mention that because we require information about the connections between foreshock structures and jets, we must use instantaneous boundaries, as otherwise the connections might be lost in time-averaging. We mention that time-averaging as done in Ng et al. [2022] is also a valid approach.**

- **We have revised the Model and Methods section to mention that because the density criterion for identifying the bow shock frequently identifies foreshock structures approaching the bow shock as consisting of magnetosheath plasma, we show it only as a comparison to the core heating and magnetosonic Mach number criteria that are actually used for the analysis.**
* * *
*In the last paragraph of Section 2.1, the authors do not provide enough physical and numerical information about the setup of 2D Vlasiator simulations they analyze in this manuscript. At the very least, they should provide the (1) domain size in solar wind proton inertial lengths, (2) mesh resolution (numbers of cells in each dimension), and (3) simulation magnetopause to obstacle radius ratio (which characterizes the simulation dipole strength chosen). For the reader to understand how realistic these simulations are, the authors should also discuss how they scale their ion inertial length scales to RE distances shown in the Figures.*

Vlasiator uses an unscaled dipole, and the values of the plasma and magnetic field properties in the simulation are in SI units.

We will add this information to Section 2.1 along with descriptions of the domain sizes. The real space and velocity space resolutions can be found in the caption of Table 1. For reference, the real space resolution will also be reported in units of $d_i$ in the solar wind.

- **We have revised the Model and Methods section to clarify that Vlasiator uses an unscaled dipole.**

- **We have added information about the domain size and extents of the simulations, in $R_{\mathrm{E}}$ as well as (solar wind) $d_i$, to the Model and Methods section.**

- **We have revised the Model and Methods section to mention the approximate standoff distance of the magnetopause and why it is slightly different from what is expected in reality.**
* * *
*Lines 160-165: It is not clear what the authors did here. VSCs are immobile points, aren't they? If so, their frame of reference coincides with the simulation frame of reference. Therefore, it is unclear what $v_{\mathrm{SC}}$ ("the propagation velocity in the spacecraft frame") represents in this numerical analysis. Does it make sense to measure it with respect to any point in the simulation (e.g., VSCs)? Isn't it just $v_n$? Please revise this paragraph for clarity.*

The references to a "spacecraft frame" are because the timing method used was developed for use with observations by real spacecraft. In our study the VSCs are indeed stationary and thus the spacecraft frame is identical to the simulation frame. We will change $v_{\mathrm{SC}}$ to $v$ to hopefully clear up any confusion. The reason why $v_n$ on its own is not used, but instead a "corrected" velocity is calculated, is that $v_n$ does not account for the

plasma bulk velocity perpendicular to the propagation direction of the assumed plane wave. The corrected velocity $v$ does take it into account.

- **We have revised the manuscript to refer to the corrected velocity as $v$ instead of $v_{\text{SC}}$ and changed mentions of "spacecraft frame" to "simulation frame" in order to be more clear.**
* * *
*Section 5 (Conclusions) needs to be revised in accordance with my previous remarks, especially those regarding "the formation mechanism proposed by Karlsson et al (2015)", who simply hypothesized about the origin of jets. There is no so evidence in this manuscript that the magnetosheath structures discussed are the SLAMS discussed by Karlsson et al (2015). The conclusion should also make clear which "jets" the authors refer to, given the nomenclature adopted in this manuscript (e.g., FCS-jets vs non-FCS jets). In regard to the simulation jets, the Conclusion mentions: "These properties indicate that they might form behind a part of the bow shock that locally and temporarily changes from quasi-parallel to quasi-perpendicular". This is not consistent with the SLAMS related explanation pointed out by Karlsson et al (2015). The statements in the Abstract and Conclusions sections must be clear and consistent.*

The sentences "These properties indicate that they might form behind a part of the bow shock that locally and temporarily changes from quasi-parallel to quasi-perpendicular" refers to the proposed explanation of the formation of flankward jets only. While this local change of the bow shock is proposed to be related to foreshock ULF waves, these ULF waves are not necessarily compressive in the way required for the formation mechanism found in Raptis et al. [2022b].

As mentioned above, we will revise the manuscript to de-emphasise how the Karlsson et al. [2015] SLAMS hypothesis relates to what we see in our simulations and to emphasise the results of Suni et al. [2021] and Raptis et al. [2022b] instead.

- **We have revised the Conclusions section to refer to Raptis et al. [2022b] instead of Karlsson et al. [2015] in the context of formation mechanisms.**
* * *
*The Abstract contains many insignificant details which obscure the main results of this work. The Abstract also contains information that must be supported by references, for instance: "A jet generation mechanism that has been widely discussed in observational and numerical studies is steepened Ultra Low Frequency (ULF) waves interacting with the bow shock. However, other formation mechanisms have also been proposed". For clarity, such statements should be avoided in the Abstract.*

We will revise the abstract to remove statements requiring references.

- **We have revised the abstract to remove statements requiring references.**
* * *
*Line 44: replace "is host" by "is a host"*

We will reformulate this sentence to "The dynamic quasi-parallel magnetosheath exhibits many kinds of transient phenomena".

- **We have reformulated the sentence accordingly.**
* * *
*Line 82: replace "E.g." by "For instance,"*

We will implement this change as requested.

- **We have implemented the change as requested.**
* * *
*Line 87: replace "simulation runs" by "two-dimensional simulation runs" (it is important to emphasize 2D at the very beginning to avoid further confusion)*

We will implement this change as requested.

- **We have implemented the change as requested.**
* * *
*Last two sentences in Section 1. It is not clear what types of "jets" are being studied in this manuscript and what makes "non-FCS jets" different from "FCS jets". This must be discussed before the reader gets to see results from the statistical analysis. Also, this manuscript lacks comparisons with observations. That must be mentioned/explained.*

In Suni et al. [2021], we found that FCS-jets have different properties (at least in terms of magnetosheath penetration depth) than non-FCS-jets, but we did not study their differences in detail and did not conclude that there must be a fundamental difference between FCS-jets and non-FCS-jets. In the present paper, we find that there is no fundamental difference between the properties of antisunward non-FCS-jets and FCS-jets. The manuscript will be revised to de-emphasise any implied fundamental differences between FCS-jets and non-FCS-jets and clarify that categorisation based on connection with foreshock structures is just one of many possible ways of categorising jets.

The lack of comparisons with observations is due to the fact that categorising jets based on propagation direction requires multiple points of observation for the timing analysis, which in the case of spacecraft observations would mean spacecraft constellations in tight formation. Combined with the difficulty of ascertaining through spacecraft observations whether a jet forms at the bow shock or not means that suitable spacecraft observations to compare to are very rare. It should however be noted that Palmroth et al. [2021] did compare jets found in the very same simulations studied here to MMS observations, finding that they are quite similar in their properties. The manuscript will be revised to explain the lack of comparisons with spacecraft observations and that previous studies have found jets in Vlasiator to be realistic.

- **We have revised the manuscript to clarify what is meant by FCS-jets and non-FCS-jets, how we came to this original categorisation, and how antisunward jets are linked to FCS-jets.**
* * *
*Line 108: mention that you are using GSM axes*

The simulation coordinate system is GSE, but because our simulation runs have a dipole that is aligned with the GSE $z$-axis, it is equivalent to GSM in this case. The lack of dipole tilt and the relation between GSM and GSE will be mentioned in the revised manuscript.

- **We have revised the Model and Methods section to mention that we use GSE coordinates, and that this coordinate system is equivalent to GSM in the case of the simulations we are studying.**
* * *
*Line 129: provide a definition of "magnetosonic Mach number" to avoid confusion. Are you using the magnetosonic speed, $v_{ms} = \sqrt{v_a^2 + v_s^2}$ so that $M_{ms} = v_{sw}/v_{ms}$?*

This is indeed the definition we are using. The definition will be explicitly mentioned in the revised manuscript.

- **We have revised the Model and Methods section to explicitly write the definition of the magnetosonic speed.**
* * *
*Line 131: rephrase "the position space simulation cells" (what does it mean?)*

By "position space" we mean the 3-dimensional space in GSE $x, y, z$ coordinates. We will reformulate this to "the simulation cells in position space"

- **We have reformulated the sentence accordingly.**

**Author response to referee #2**

The authors wish to thank Anonymous referee #2 for their very insightful and thorough comments on the manuscript. We will take the comments into account when revising the manuscript. In this document we provide responses to each of the referee's comments (formatted as italics in indented paragraphs).
* * *
*paper often refers that the results support the model by Karlsson et al 2015, but that paper states "An unambiguous answer to the question of how magnetosheath jets and plasmoids are formed is not available, and it is outside the scope of this paper to provide it." It would be very helpful to be more specific on what exactly in Karlsson et al 2015 paper the current study supports. It should be very clear what jet formation model the current study supports and clearly show evidence for that.*

Indeed, Karlsson et al. [2015] base their hypothesis of jets being caused by SLAMS on previous simulation and observational studies, as well as observed similarities between SLAMS and paramagnetic plasmoids in the magnetosheath. Because evidence in favour of this mechanism was found in both spacecraft observations [Raptis et al., 2020] and simulations Palmroth et al. [2018], we investigated the transmission of compressive foreshock structures of varying amplitudes of magnetic field enhancement through the bow shock in Suni et al. [2021] and found that a majority of the jets found in these four Vlasiator simulation runs are connected to such foreshock structures. Though the majority of these structures were found to not fulfill the SLAMS criteria, we nonetheless concluded that we had found evidence for the part of the Karlsson et al. [2015] hypothesis that concerns the transmission of structures from the foreshock to the magnetosheath. Raptis et al. [2022b] observed a connection between foreshock waves, compressive structures, bow shock reformation, and magnetosheath jet formation directly using MMS spacecraft observations, further supporting this formation mechanism. We also found that SLAMS and the weaker, more frequently forming structures that are associated with the formation of the majority of jets are part of a continuum of foreshock structures of varying magnetic field enhancement. It should also be noted that SLAMS and their role in bow shock reformation have been studied in Vlasiator simulations [Johlander et al., 2022].

We will revise the manuscript to de-emphasise the aspect of the Karlsson et al. [2015] hypothesis that specifies SLAMS as a possible origin of jets, and instead refer to the results of Suni et al. [2021] and Raptis et al. [2022b].

- **We have revised the manuscript to discuss the formation mechanism of FCS-jets and antisunward jets in the context of Suni et al. [2021] and Raptis et al. [2022b] instead of Karlsson et al. [2015].**
* * *
*It seems that the current manuscript is updating on the earlier results and shows that the previous division of jets in FCS and non-FCS type of jets is physically not appropriate. Just to remind that Suni et al 2021 was concluding that FCS and non-FCS jets have different sources, while this manuscript shows that it is not true. Therefore, it is unclear why the manuscript wants to distinguish between FCS and antisunward jets if their properties are similar. The conclusion section is written in a way that the separation between FCS and antisunward jets is not mentioned, and it would be appropriate to rewrite the whole paper also in the same way. FCS separation can be mentioned in the introduction or discussion in terms of putting in perspective the current manuscript to the earlier work but it is not appropriate from the point of view of discussing the main results of the paper. If authors want to keep FCS notations, then it has to be defined in a way that it includes antisunward jets.*

In Suni et al. [2021] we found that FCS-jets and non-FCS-jets are different in some of their properties, such as magnetosheath penetration depth, but we did not study these differences in detail and did not conclusively find that there must be fundamental differences between FCS-jets and non-FCS-jets. In this study, we find that because antisunward jets appear to form the same way as FCS-jets, the separation into FCS-jets and non-FCS-jets as defined in Suni et al. [2021] does not produce a clear separation of formation mechanisms.

We will revise the manuscript to emphasise that categorising jets based on whether they are connected to FCS or not and what way they propagate are simply two different ways of categorising jets, with the former focusing on the origins of the jets with respect to a specifically defined structure, and the latter on the properties of the jets. We will also clarify that the set of FCS-jets appears to not contain all the jets that are associated with foreshock structures of enhanced dynamic pressure. The ones not contained in the set are the antisunward non-FCS-jets, which are apparently connected to foreshock structures that are too weak to fulfill the FCS criteria as defined in Suni et al. [2021] and this study. In the discussion, we will specify that in terms of formation mechanism, antisunward non-FCS-jets and FCS-jets appear to be the same, and that the separation method in Suni et al. [2021] produced an artificial divide between them.

- **We have revised the manuscript to mention that the division of jets into antisunward and flankward arose when analysing the jets that were left by Suni et al. [2021] for a future study and thus not studied in detail there, and to emphasise that the thresholds used to define FCS-jets in Suni et al. [2021] did not identify all jets that are associated with foreshock structures of enhanced dynamic pressure. We also emphasise that antisunward jets and FCS-jets seem to have the same properties and origin.**
* * *
*As a follow-up on the previous point, the abstract is too long.*

We will shorten the abstract by removing statements that require references.

- **We have restructured the abstract to make it shorter and to remove statements that require references.**
* * *
*The methodology part requires improvements.*

We will attempt to remedy this by expanding the Model and Methods section to describe in more detail the technical aspects of the simulation. Additionally, we will elaborate on the timing analysis method in the jet classification section.

- **We have expanded the Model and Methods section by adding more technical details, and elaborating on the choices of bow shock criteria.**

- **We have revised the Results section to elaborate on the timing analysis method as well as describing how it compares to the tracking method to acquire propagation velocities.**
* * *
*What is the spacecraft frame, and how it distinguishes from other frames?*

The term "spacecraft frame" is a remnant of the fact that the timing method was developed for observations by real spacecraft. In the case of our simulations the VSCs do not move and thus the spacecraft frame is identical to the simulation frame. We will change $v_{sc}$ to $v$ in the revised manuscript to clarify this.

- **We have revised the manuscript to change $v_{SC}$ to $v$ and "spacecraft frame" to "simulation frame" (as the two are equivalent using our methodology) in order to reduce confusion.**
* * *
*What exactly is spacecraft timing done - does one times pressure increase, pressure maximum, or just correlates pressure time series? There is no example showing how well the timed series align and how good is the plane wave assumption. In simulations no nice planar wave structures are seen.*

The dynamic pressure time series are cross-correlated with each other to give time lags that are then translated into velocity based on the separation of the VSCs. Indeed, many of the jets studied in these simulations are similar in size to the VSC separation and cannot be considered planar. However, the requirement that the smallest of the maximum cross-correlation coefficients (we get one maximum for each cross-correlated pair of time-series) should be at least 0.8 (the maximum possible being 1) ensures that the dynamic pressure enhancements observed by different VSCs are similar to each other in shape and environment and are thus most likely part of the same structure, giving a reasonable estimate of the propagation velocity even though the structures are not planar. We will explain this in the revised manuscript.

- **We have revised to Results section to elaborate on how the timing analysis is performed and how we ensure that results of timing analyses with too low cross-correlation are not included in the statistical analysis.**
* * *
*There is no discussion or explanation of what is the physical meaning of estimating vsc and vtr, and what are the expected differences.*

$v_{sc}$, now renamed $v$, is derived from observations of dynamic pressure enhancements as temporal structures at a limited number of stationary points in space, which is similar to how spacecraft constellations observe in reality.

$v_{tr}$, on the other hand, is based on the motion of a spatial structure defined as a collection of spatially connected simulation cells whose dynamic pressures exceed the jet threshold.

These velocities are expected to be virtually identical in situations where the dynamic pressure enhancement is spatially large enough to be observed by all three VSCs and its shape is not complex and does not change significantly as it propagates. If the enhancement is spatially very small (on the order of 1 simulation cell), the fact that the tracking method considers only cells whose dynamic pressure fulfills the jet criteria while the timing method considers any dynamic pressure enhancement can lead to differences in the velocities.

The observation that the median $v_{tr}$ is smaller than the median $v$ in Figure 2 in the manuscript is probably due to the elongation of jets after they form at the bow shock, causing a significant change in shape and thus a difference in the propagation of the weighted center of the jet ($v_{tr}$) and the peak of the dynamic pressure enhancement (which is what $v$ is based on).

We will explain this in the revised manuscript.

- **We have revised the Results section to explain the physical meanings of the two different propagation velocities, and when we expect differences or agreement between the two velocities.**
* * *
*Recent studies, such as Raptis et al 2022, show that simply looking on moments is not enough, it can be too simplified and misleading picture in understanding jets as plasma can contain different populations in the same location. Vlasov simulations have the great advantage of having the full distribution function and allowing to resolve the different populations in distribution functions. At least for the case studies it would be important to include the distribution functions of jets, to support the usage of moments.*

While our simulations have VDFs in every spatial cell during runtime, it is unfortunately unfeasible to save all VDFs in the output data files as this would increase the file size from $\sim$2 GB to $\sim$2 TB. Due to this, VDFs are saved for only certain cells. Of the non-FCS-jets investigated in this study, only one happens to overlap with a cell whose VDF is stored. We have analysed this VDF and found that indeed, slightly after the jet starts overlapping with the VDF-carrying cell, which is at the bow shock, the reduced 1D distributions show evidence of two maxima in the plasma distribution function. Some FCS-jets overlapped with VDF-carrying cells deeper in the magnetosheath. In these cases, the reduced 1D distributions show bumps on their tails, but at no point did we see unconnected populations in the distribution functions. However, studying the possible effects of multiple populations on moment calculations is outside the scope of this study.

We will revise the manuscript to mention that these multiple populations could be an issue when calculating moments under the assumption of a single population, with a reference to Raptis et al. [2022a].

- **We have revised the Discussion section to mention the potential effect of multiple populations on the study of magnetosheath jets via plasma moments only.**
* * *
*l.13 propagate > propagating*

We think "propagate" is correct here.
* * *
*l.156 It is confusing to have VSC as a notation for virtual s/c as later one uses also $v_{sc}$ for the velocity of the dynamic pressure pulse.*

We will change $v_{sc}$ to $v$, for the reasons described above, which should hopefully clear up any confusion.

- **We have changed $v_{sc}$ to $v$.**
* * *
*Figure 5: red, orange, and pink are too close colors to be distinguishable. Valid also in other figures.*

We will change the colormaps and contour colours in figures 1,3,4, and 5 for better visual clarity.

- **We have changed the colourmaps and contour colors in the mentioned figures for better visual clarity.**

**References**

H. Hietala, T. V. Laitinen, K. Andréeová, R. Vainio, A. Vaivads, M. Palmroth, T. I. Pulkkinen, H. E. J. Koskinen, E. A. Lucek, and H. Rème. Supermagnetosonic Jets behind a Collisionless Quasiparallel Shock. *Physical Review Letters*, 103(24):245001, December 2009. doi: 10.1103/PhysRevLett.103.245001.

A. Johlander, M. Battarbee, L. Turc, U. Ganse, Y. Pfau-Kempf, M. Grandin, J. Suni, V. Tarvus, M. Bussov, H. Zhou, M. Alho, M. Dubart, H. George, K. Papadakis, and M. Palmroth. Quasi-parallel Shock Reformation Seen by Magnetospheric Multiscale and Ion-kinetic Simulations. *Geophysical Research Letters*, January 2022. ISSN 0094-8276, 1944-8007. doi: 10.1029/2021GL096335.

T. Karlsson, A. Kullen, E. Liljeblad, N. Brenning, H. Nilsson, H. Gunell, and M. Hamrin. On the origin of magnetosheath plasmoids and their relation to magnetosheath jets. *Journal of Geophysical Research: Space Physics*, 120(9):7390–7403, 2015. ISSN 2169-9402. doi: 10.1002/2015JA021487.

J. Ng, L.-J. Chen, Y. Omelchenko, Y. Zou, and B. Lavraud. Hybrid Simulations of the Cusp and Dayside Magnetosheath Dynamics Under Quasi-Radial Interplanetary Magnetic Fields. *Journal of Geophysical Research: Space Physics*, 127(10), October 2022. ISSN 2169-9380, 2169-9402. doi: 10.1029/2022JA030359.

Y. A. Omelchenko, L.-J. Chen, and J. Ng. 3D Space-Time Adaptive Hybrid Simulations of Magnetosheath High-Speed Jets. *Journal of Geophysical Research: Space Physics*, n/a(n/a):e2020JA029035, 2021. ISSN 2169-9402. doi: 10.1029/2020JA029035.

M. Palmroth, M. Archer, R. Vainio, H. Hietala, Y. Pfau-Kempf, S. Hoilijoki, O. Hannuksela, U. Ganse, A. Sandroos, S. Von Alfthan, and J. P. Eastwood. ULF foreshock under radial IMF: THEMIS observations and global kinetic simulation Vlasiator results compared: ULF WAVES IN THE RADIAL FORESHOCK. *Journal of Geophysical Research: Space Physics*, 120(10):8782–8798, October 2015. ISSN 21699380. doi: 10.1002/2015JA021526.

Minna Palmroth, Heli Hietala, Ferdinand Plaschke, Martin Archer, Tomas Karlsson, Xochitl Blanco-Cano, David G. Sibeck, Primoz Kajdic, P. Kajdic, Urs Ganse, Yann Pfau-Kempf, Markus Battarbee, and Lucile Turc. Magnetosheath jet properties and evolution as determined by a global hybrid-Vlasov simulation. *Annales Geophysicae*, 36(5):1171–1182, September 2018. doi: 10.5194/angeo-36-1171-2018.

Minna Palmroth, Savvas Raptis, Jonas Suni, Tomas Karlsson, Lucile Turc, Andreas Johlander, Urs Ganse, Yann Pfau-Kempf, Xochitl Blanco-Cano, Mojtaba Akhavan-Tafti, Markus Battarbee, Maxime Dubart, Maxime Grandin, Vertti Tarvus, and Adnane Osmane. Magnetosheath jet evolution as a function of lifetime: Global hybrid-Vlasov simulations compared to MMS observations. *Annales Geophysicae*, 39(2):289–308, March 2021. ISSN 0992-7689. doi: 10.5194/angeo-39-289-2021.

Savvas Raptis, Tomas Karlsson, Ferdinand Plaschke, Anita Kullen, and Per-Arne Lindqvist. Classifying Magnetosheath Jets Using MMS: Statistical Properties. *Journal of Geophysical Research-Space Physics*, 125(11):e2019JA027754, November 2020. ISSN 2169-9380. doi: 10.1029/2019JA027754.

Savvas Raptis, Tomas Karlsson, Andris Vaivads, Martin Lindberg, Andreas Johlander, and Henriette Trollvik. On Magnetosheath Jet Kinetic Structure and Plasma Properties. *Geophysical Research Letters*, 49(21): e2022GL100678, 2022a. ISSN 1944-8007. doi: 10.1029/2022GL100678.

Savvas Raptis, Tomas Karlsson, Andris Vaivads, Craig Pollock, Ferdinand Plaschke, Andreas Johlander, Henriette Trollvik, and Per-Arne Lindqvist. Downstream high-speed plasma jet generation as a direct consequence of shock reformation. *Nature Communications*, 13(1):598, December 2022b. ISSN 2041-1723. doi: 10.1038/s41467-022-28110-4.

J. Suni, M. Palmroth, L. Turc, M. Battarbee, A. Johlander, V. Tarvus, M. Alho, M. Bussov, M. Dubart, U. Ganse, M. Grandin, K. Horaites, T. Manglayev, K. Papadakis, Y. Pfau-Kempf, and H. Zhou. Connection Between Foreshock Structures and the Generation of Magnetosheath Jets: Vlasiator Results. *Geophysical Research Letters*, 48(20), October 2021. ISSN 0094-8276, 1944-8007. doi: 10.1029/2021GL095655.

L. Turc, O. W. Roberts, D. Verscharen, A. P. Dimmock, P. Kajdič, M. Palmroth, Y. Pfau-Kempf, A. Johlander, M. Dubart, E. K. J. Kilpua, J. Soucek, K. Takahashi, N. Takahashi, M. Battarbee, and U. Ganse. Transmission of foreshock waves through Earth's bow shock. *Nature Physics*, 19(1):78–86, January 2023. ISSN 1745-2473, 1745-2481. doi: 10.1038/s41567-022-01837-z.

---

## Referee Report (RR1)

The manuscript by Suni et al. (2023) utilizes Vlasiator hybrid-Vlasov simulations to study foreshock compressive structures (FCSs) as a cause of downstream dynamic pressure enhancements ("magnetosheath jets"). This work builds on previous results by Suni et al. (2021), where 75 % of jets were found to be associated with FCSs crossing the shock. Here the authors focus on the last 25 % of jets, which they show can be divided into anti-sunward and flankward jets. The authors show that the anti-sunward jets are related to weaker foreshock compressive structures. The manuscript is an important contribution towards understanding magnetosheath jet formation. The manuscript is well-written and the methods are clearly described.

I have only minor comments/questions regarding the interpretation of the results and a few suggestions for improving the text. I believe the manuscript will be suitable for publication after minor revisions.

**Main comments:**

1. The authors refer to Suni et al. (2021) as "the previous study" multiple times in the abstract, so a citation should be included. I think the authors should clearly mention in the beginning that this study concerns jets that were not associated to FCSs in Suni et al. (2021). Currently, this information is dispersed around the abstract: "We focus on jets whose origins have not been clearly determined in a previous study using the same simulations" and "from those of jets found in a previous study, which were associated with foreshock structures of enhanced dynamic pressure and magnetic field".

2. The authors should discuss the nature of these foreshock compressive structures more in the manuscript. How do these FCSs move with respect to the solar wind flow? Do these structures appear locally or do they travel from far upstream? Which FCSs produce jets and which do not? I understand that answering some of these questions may require further analysis, and I do not think that is required. However, as the nature of FCSs is an integral part of the jet formation mechanism being suggested, it should be discussed more.

3. In Figure 5 it seems like in all of the panels and for all shock bow shock criteria, the shock is initially closer to the Earth than after the jet. Could this be a signature of a local corrugation in the shock? Or is it simply due to the bow shock standoff distance increasing as a function of time in the simulation?

4. It is not clear to me how the results agree with the exact mechanism suggested by Raptis et al. (2022), more so than the one hypothesized by Karlsson et al. (2015). In the simulations, are the FCSs reforming the shock and jets forming as solar wind is compressed between the old and the new shock or are the FCSs themselves crossing the shock and emerging downstream as jets? In the Discussion and Conclusions sections, the authors should be more specific about how their results support the mechanism suggested by Raptis et al. (2022).

**Minor suggestions:**

Line 2: "dynamics pressure" to "dynamic pressure".

Lines 52-54: This sentence is unclear to me. What was the definition deemed most appropriate for? If you mean that this definition was found to be the most appropriate one for the goal of capturing transient enhancements in dynamic pressure, I suggest rewording the sentence.

---

## Author Response (AR2)

**Author response**

The authors wish to thank the referees who reviewed the revised manuscript. Their comments and feedback have contributed significantly to improving the quality and structure of the manuscript.

**Author response to referee #3**

In this document we provide responses to each of the referee's comments (formatted as italics in indented paragraphs).

> *The authors refer to Suni et al. (2021) as "the previous study" multiple times in the abstract, so a citation should be included. I think the authors should clearly mention in the beginning that this study concerns jets that were not associated to FCSs in Suni et al. (2021). Currently, this information is dispersed around the abstract: "We focus on jets whose origins have not been clearly determined in a previous study using the same simulations" and "from those of jets found in a previous study, which were associated with foreshock structures of enhanced dynamic pressure and magnetic field"*

We have added a reference to Suni et al. [2021] in the abstract, and compiled the information about the nature of the FCS-jets in the sentence where the previous study is first mentioned.

> *The authors should discuss the nature of these foreshock compressive structures more in the manuscript. How do these FCSs move with respect to the solar wind flow? Do these structures appear locally or do they travel from far upstream? Which FCSs produce jets and which do not? I understand that answering some of these questions may require further analysis, and I do not think that is required. However, as the nature of FCSs is an integral part of the jet formation mechanism being suggested, it should be discussed more.*

We have not investigated the nature and formation of FCSs in detail, but we have added an introduction to shocklets and SLAMS, which are partially a subset of FCSs, to the introduction section.

> *In Figure 5 it seems like in all of the panels and for all shock bow shock criteria, the shock is initially closer to the Earth than after the jet. Could this be a signature of a local corrugation in the shock? Or is it simply due to the bow shock standoff distance increasing as a function of time in the simulation?*

We believe that this outward motion of the bow shock criteria is mostly due to the increasing bow shock standoff distance, but the short-lived perturbation around the formation time of the jets could be local corrugation. This is now mentioned in the manuscript.

> *It is not clear to me how the results agree with the exact mechanism suggested by Raptis et al. (2022), more so than the one hypothesized by Karlsson et al. (2015). In the simulations, are the FCSs reforming the shock and jets forming as solar wind is compressed between the old and the new shock or are the FCSs themselves crossing the shock and emerging downstream as jets? In the Discussion and Conclusions sections, the authors should be more specific about how their results support the mechanism suggested by Raptis et al. (2022).*

We have added a clarification, referring to Fig. 5, that we observe increasing density and magnetic field with decreasing distance from the bow shock in the foreshock structures that are associated with antisunward and FCS-jets, and that this agrees with the steepening of the foreshock waves that are associated with bow shock reformation and the jet in Raptis et al. [2022].

*Line 2: "dynamics pressure" to "dynamic pressure".*

We have corrected the spelling as requested.

*Lines 52-54: This sentence is unclear to me. What was the definition deemed most appropriate for? If you mean that this definition was found to be the most appropriate one for the goal of capturing transient enhancements in dynamic pressure, I suggest rewording the sentence.*

We have clarified that we found the definition in question to be most appropriate for capturing transient enhancements of dynamic pressure.

**References**

Savvas Raptis, Tomas Karlsson, Andris Vaivads, Craig Pollock, Ferdinand Plaschke, Andreas Johlander, Henriette Trollvik, and Per-Arne Lindqvist. Downstream high-speed plasma jet generation as a direct consequence of shock reformation. *Nature Communications*, 13(1):598, December 2022. ISSN 2041-1723. doi: 10.1038/s41467-022-28110-4.

J. Suni, M. Palmroth, L. Turc, M. Battarbee, A. Johlander, V. Tarvus, M. Alho, M. Bussov, M. Dubart, U. Ganse, M. Grandin, K. Horaites, T. Manglayev, K. Papadakis, Y. Pfau-Kempf, and H. Zhou. Connection Between Foreshock Structures and the Generation of Magnetosheath Jets: Vlasiator Results. *Geophysical Research Letters*, 48(20), October 2021. ISSN 0094-8276, 1944-8007. doi: 10.1029/2021GL095655.